# A full year of continuous net soil and ditch $CO_2$, $CH_4$, $N_2O$ fluxes, soil hydrology and meteorology for a drained fen in Denmark

Annelie S. Nielsen[1], Klaus S. Larsen[1], Poul Erik Lærke[2], Andres F. Rodriguez[2], Johannes W.M. Pullens[2], Rasmus J. Petersen[3], Jesper R. Christiansen[1]

[1]Department of Geoscience and Natural Resource Management, University of Copenhagen, Frederiksberg, DK-2000, Denmark
[2]Department of Agroecology, Aarhus University, Tjele, DK-8830, Denmark
[3]Department of Ecoscience, Aarhus University, Aarhus, DK-8000, Denmark

*Correspondence to*: Jesper R. Christiansen (jrc@ign.ku.dk)

**Abstract.** We present a detailed dataset (https://doi.org/10.60612/DATADK/BZQ8JE) of automated greenhouse gas (GHG) net soil and ditch fluxes of carbon dioxide ($CO_2$), methane ($CH_4$), and nitrous oxide ($N_2O$) from a drained fen in Denmark covering a full year. The dataset resolves small scale spatial and hourly-daily-seasonal dynamics of GHG soil fluxes. The GHG flux dataset is accompanied by simultaneous time series of soil temperature and moisture, as well as groundwater table depth and covers spatiotemporal gradients in soil hydrological and climatic variability. The GHG fluxes of $CO_2$, $CH_4$ and $N_2O$ were measured simultaneously by a high-precision cavity ring down laser spectrometer connected with a novel automated GHG system platform called SkyLine2D (Earthbound Scientific Ltd., UK) that allowed up to 27 individual chamber measurement points along a 24 meter transect. In total 47.483 chamber measurements were completed and after quality control 44.631 $CO_2$ fluxes, 44.099 $N_2O$ and 42.515 $CH_4$ fluxes remained.

The average (±SE) net soil $CO_2$ efflux observed at the site ($2.6\pm0.02$ µmol $CO_2$ m$^{-2}$ s$^{-1}$ or $35\pm0.3$ t$CO_2$ ha$^{-1}$ y$^{-1}$) aligns with findings from similar drained fens in northern Europe covering substantial spatial variability. The organic soil at the site was a larger net source of $N_2O$ ($8.9\pm0.1$ nmol $N_2O$ m$^{-2}$ s$^{-1}$ or $123\pm1.4$ kg $N_2O$ m$^{-2}$ ha$^{-1}$ y$^{-1}$) to the atmosphere compared to other temperate drained organic grassland soils in northern Europe with similar spatial variability as soil $CO_2$ effluxes. However, the temporal variability of $N_2O$ fluxes were closely linked to fluctuations of the groundwater table depth with emission bursts of soil $N_2O$ emissions during low water table depth. $N_2O$ fluxes decreased to near-zero fluxes when the water table depth increased. Net soil $CH_4$ fluxes were near-zero and the site overall acted as a smaller net source ($0.18\pm0.06$ nmol $CH_4$ m$^{-2}$ s$^{-1}$ or $0.91\pm0.3$ kg $CH_4$ ha-1 y-1) compared to other drained organic grassland soils, although net uptake of atmospheric $CH_4$ was observed as well especially in drier conditions. Compared to the peat soil, the ditch was a smaller net source of $CO_2$ ($0.94\pm0.05$ µmol $CO_2$ m$^{-2}$ s$^{-1}$ or $1.3\pm0.7$ t$CO_2$ ha$^{-1}$ y$^{-1}$) and $N_2O$ ($0.35\pm0.03$ nmol $N_2O$ m$^{-2}$ s$^{-1}$ or $4.9\pm0.4$ kg $N_2O$ ha$^{-1}$ y$^{-1}$). The ditch emission of $CH_4$ ($161\pm13$ nmol $CH_4$ m$^{-2}$ s$^{-1}$ or $812\pm66$ kg $CH_4$ ha$^{-1}$ y$^{-1}$) average of diffusive and ebullition fluxes) to the atmosphere was more than two orders of magnitude larger than net the soil $CH_4$ emissions.

The very large number of fluxes of $CO_2$, $N_2O$ and $CH_4$ for peat soils and a ditch linked to both groundwater table data, soil moisture/temperature as well as groundwater and soil physicochemical parameters are unique to northern temperate peatlands and holds a potential for exploring and testing basic hypothesis on the simultaneous regulation of these gas fluxes by both soil hydrology and temperature, including soil and

groundwater chemistry. The high temporal detail also allows for time series analyses as well as investigations
into diurnal and seasonal patterns of fluxes in response to physical drivers. Similarly, the high frequency of
measured variables and the large number of spatial replicates are furthermore well suited for testing
biogeochemical models as it is possible to have both calibration and validation dataset covering the same period.
Furthermore, the surprisingly large spatial variability of flux data is ideal to include in model sensitivity tests
which can aid in constraining model outputs and develop model routines.

## 1 Introduction

Understanding the climate feedback of temperate drained and rewetted wetlands requires robust observational datasets of net fluxes, e.g. whether the rewetted peatlands act as net sources to the atmosphere or sinks of greenhouse gases (GHG). This necessitates being able to capture spatial and temporal variability from these systems. Flux data covering all three major GHGs are rare for temperate peatlands, and despite growing efforts to quantify GHG fluxes from drained peatlands, existing datasets often suffer from limited temporal resolution, short monitoring periods, or a lack of concurrent hydrological and meteorological data. Many studies rely on manual chamber-based campaigns that may be able to capture overall seasonal dynamics, but fail to capture short term transient emission phenomenon in response to fluctuations in physical drivers, for example fluctuating groundwater. Also, manual based measurements are labour intensive limiting the number of spatial replicates. Moreover, current high temporal resolution datasets for wetlands using eddy covariance typically offer high good quality for a specific wetland site, but it is challenging to derive the specific spatial variability across the different sub-environments within the wetland, for example between hummocks and hollows with different GHG emission profiles. This discrepancy between spatial and temporal coverage of current flux methodologies in wetlands in turn hampers the ability to develop precise models that integrate spatiotemporal patterns and can forecast GHG fluxes at the ecosystem scale more precisely. This can impact the ability to predict climatic feedback of wetlands now and under future alteration of these systems driven by land use and climatic changes.

However, automated GHG closed chamber flux measurements from ecosystems are becoming increasingly common, also in peatland research (Anthony and Silver, 2023; Boonman et al., 2024) as equipment costs decrease and awareness grows about the importance of resolving temporal variability of GHG fluxes to better understand soil biogeochemical processes and soil-climate feedback. But high-frequency data of GHG fluxes are still scarce for peatlands and spatial variability of fluxes is rarely represented as well due to limited number of spatial replicates. Thus, most automated chamber systems are setup around a multiplexer control unit linking multiple chambers with one or more GHG analysers. State-of-the-art automatic chamber systems, like the LI-8250 Automated Gas Flux System (LiCOR, USA) or the eosAC-LT/LO (Eosense Inc. Canada), i.e. allow for a standard number of 8 or 16 chambers, respectively, that can be upgraded to 36 chambers with additional manifolds. Such large replicate chambers allow for improved characterization of spatial variation or treatment effects coupled with temporal variations but are costly to establish.

We here present a dataset that addresses the abovementioned limitations by combining high-frequency, continuous measurements of net soil fluxes of carbon dioxide ($CO_2$), methane ($CH_4$) and nitrous oxide ($N_2O$) with detailed hydrological and meteorological variables. The GHG fluxes were measured with an automated GHG, called SkyLine2D, chamber system over 12 months resolving spatiotemporal patterns of GHG fluxes including 27 individual collars (26 on organic soil and 1 in a ditch) over a 24 m transect on a temperate drained fen peatland. Integrated quality control, flagging of erroneous or uncertain flux measurements enabled objective filtering of poor-quality data on the entire dataset. This comprehensive spatiotemporal coverage enables robust calibration and validation of biogeochemical and hydrological models, particularly those aiming to simulate the complex interactions between water table dynamics, soil processes, and GHG emissions in managed peatland systems.

Considering the critical need for obtaining high-quality data on soil GHG fluxes from natural and restored
peatlands in Europe and globally, our dataset marks an important contribution to this endeavour as it addresses
current data shortcomings for Danish and European peatlands by providing detailed data on temporal and spatial
patterns of GHG fluxes from organic soils and drainage ditches together with environmental drivers of soil
hydrology and temperature, organic soil properties and groundwater geochemistry. We publish this data with the
aim of it being used by the scientific community for both experimentalists to test hypothesis of how GHG
dynamics are related to hydrology, soil, geochemistry and climate, as well as for the modelers to test and
develop biogeochemical models for peat lands.
## 2 Materials and Methods
### 2.1 Site description
The field site, Vejrumbro (N 56.43819 E 9.54527 (WGS 84)), is located in Central Jutland, in Denmark near the
city of Viborg (Fig. 1A) with a mean annual temperature of 8.3°C and annual precipitation of 675 mm for the
period 1991–2020 (measured 6 km away at Aarhus University Viborg Meteorological Station in Foulum
(Jørgensen et al., 2023)). It is situated in the Nørre Å valley and is characterized as a riparian fen peat soil (Reza
Mashhadi et al., 2024). The riparian fen developed in a former glacial river valley with flat topography gently
sloping (<2.5 meters over 300 meters) towards the Nørre Å that forms the central river in this area (Fig. S1).
The site was drained in 1950 with ditches and tile drains for cultivation and was used to cut hay for fodder as the
conditions were unfavourable for cereal production (Nielsen et al., 2024). Since 2018, Vejrumbro has been a
living lab for agroecological research managed by the Department of Agroecology at Aarhus University. From
2018, the site had a passive rewetting strategy by terminating maintenance of the open ditches. During 2022, the
main ditches were gradually blocked.
2.1.1 Site preparation
We chose to perform the flux measurements without aboveground plants as the small chamber dimensions
(height of 20 cm) prohibited inclusion of these in the chamber as the plants typically reach over 100 cm in
height at this site. The strategy was therefore to focus on measuring net soil GHG fluxes, where we assume the
contribution of gases are derived from heterotrophic respiration of older peat C/N, root exudated C/N from
adjacent plants, dissolved N in groundwater and belowground autotrophic respiration ($CO_2$) from roots
inhabiting the peat below the collars. We are aware that omitting plants prohibit a full evaluation of the net
ecosystem exchange of GHG and hence its net climate impact, as the aboveground plants represent a net sink of
atmospheric $CO_2$ and can increase the emission of $CH_4$ and $N_2O$ (Jørgensen et al., 2012; Vroom et al., 2022).
However, by removing plants we isolate the soil processes leading to net soil emission/uptake of the GHG.
Collectively, this can provide a mechanistic insight into the regulation of fluxes by hydrology and temperature.
We acknowledge that studies of GHG fluxes in peatlands should seek to include the aboveground plant
component to the net GHG flux from the ecosystem if possible.
Two months prior to collar installation in summer 2021, we cleared vegetation within and around each collar
(~40 × 40 cm) by harvesting and applying a single recommended dose of glyphosate (~100 mg m$^{-2}$) to
aboveground plants only, avoiding soil contact. Glyphosate's average half-life in mineral soils is ~21 days,

ranging from 6–87 days and increasing with clay content (Padilla and Selim, 2020). Given the low dose and absence of clay, residual glyphosate was likely minimal during flux measurements. Although repeated applications can suppress microbial activity (Nguyen et al., 2016), the single treatment months prior suggests limited direct impact on microbial respiration. Still, transient effects cannot be ruled out, and the lack of an untreated control prevents quantification. Regrowth inside collars was manually removed at least weekly, minimizing photosynthetic $CO_2$ uptake. While regrowth abundance was not measured, stable net $CO_2$ efflux between removals suggests minimal impact. Aboveground plant removal is standard for isolating soil GHG fluxes, though belowground autotrophic respiration from adjacent roots remained, as trenching was avoided to reduce site disturbance. Without a control plot, the direct effect of disturbance on GHG fluxes remains uncertain.

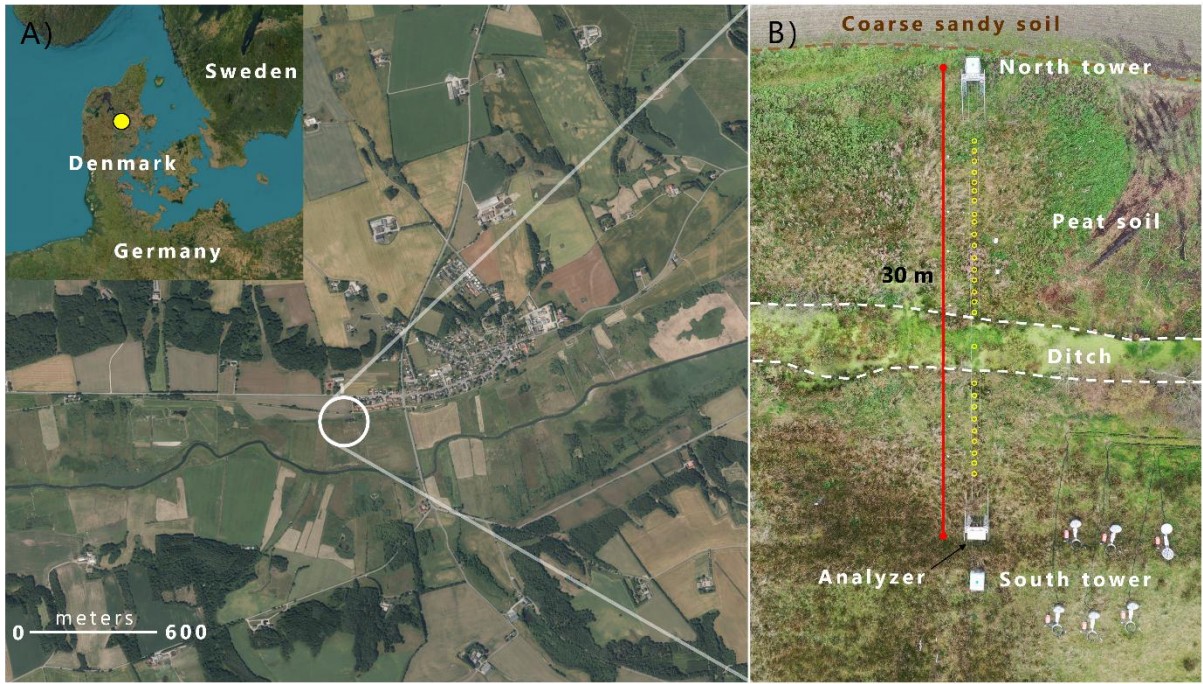

**Figure 1: A) The Vejrumbro location in Jutland (N 56.43819 E 9.54527 (WGS 84)) in the Nørre Å valley near the village of Vejrumbro. The grey circle marks the placement of the SkyLine2D system. Satellite images: © Google Earth. B) Drone image of the measurement transect (September 27th, 2023) after flux measurements had stopped. Dashed brown line marks the approximate boundary between the agricultural field, coarse sandy soil (north) and the peat/organic soil (south). The red line marks the end points of the SkyLine2D system (30 meters). The open yellow circles (n=27) mark the approximate position of individual collars across the transect of the field (24 meters in length) where greenhouse gas fluxes were measured. The ditch is located between the dashed white lines. The analyser was placed at the south tower. Elevation above sea level along the 24-meter collar transect varied from 3.77 m in the south to 4.06 m in the north.**

## 2.2 Overview of time series of GHG fluxes, soil temperature/moisture, air temperature, wind direction and groundwater level

The dataset is comprised of a 12-month time series of net soil fluxes of $CO_2$, $CH_4$ and $N_2O$, accompanied by a longer timeseries of soil temperature and moisture at 5 cm depth, meteorological variables (air temperature, wind speed and direction measured at 2 meter height) and a shorter time series groundwater table level, depth and temperature (Table 1). Due to equipment failure of the SkyLine2D the GHG flux measurements started on

February 2nd, 2022 and ended January 28th, 2023 in total 360 days (Table 1). Groundwater level measurements
started between March 9th to 31st, 2022 (Table 1). All other variables were measured continuously from July 1st,
2021, until January 31st, 2023 (Table 1). In the period between December 7th and 19th, 2022 intermittent periods
of snow cover (depth was not measured) on the ground occurred. This snow cover did not impede flux
measurements.

**Table 1: Available time series data from the Vejrumbro SkyLine2D system. Coloured Data availability for each variable in 2021 to 2023 is indicated with horizontal bars.**

| Variable | Unit | Model/sensor type | Frequency (minutes) | Aug 2021 | Sep | Oct | Nov | Dec | Jan 2022 | Feb | Mar | Apr | May | Jun | Jul | Aug | Sep | Oct | Nov | Dec | Jan 2023 |
|---|---|---|---|---|---|---|---|---|---|---|---|---|---|---|---|---|---|---|---|---|---|
| CO$_2$ flux* | μmol CO$_2$ m$^{-2}$ s$^{-1}$ | G2508 (Picarro Inc., USA) | ~10** | | | | | | | ----- | ----- | ----- | ----- | ----- | ----- | ----- | ----- | ----- | ----- | ----- | ----- |
| CH$_4$ flux* | nmol CH$_4$ m$^{-2}$ s$^{-1}$ | G2508 (Picarro Inc., USA) | ~10** | | | | | | | ----- | ----- | ----- | ----- | ----- | ----- | ----- | ----- | ----- | ----- | ----- | ----- |
| N$_2$O flux* | nmol N$_2$O m$^{-2}$ s$^{-1}$ | G2508 (Picarro Inc., USA) | ~10** | | | | | | | ----- | ----- | ----- | ----- | ----- | ----- | ----- | ----- | ----- | ----- | ----- | ----- |
| Soil temperature at 5 cm depth*** | °C | RXW-TMB-868 (Onset, USA) | 5 | ----- | ----- | ----- | ----- | ----- | ----- | ----- | ----- | ----- | ----- | ----- | ----- | ----- | ----- | ----- | ----- | ----- | ----- |
| Soil water content at 5 cm depth*** | (cm$^3$ cm$^{-3}$) | RXW-SMD-868 (5HS) (Onset, USA) | 5 | ----- | ----- | ----- | ----- | ----- | ----- | ----- | ----- | ----- | ----- | ----- | ----- | ----- | ----- | ----- | ----- | ----- | ----- |
| Air temperature at 2 m height | °C | S-THC-M002 (Onset, USA) | 5 | ----- | ----- | ----- | ----- | ----- | ----- | ----- | ----- | ----- | ----- | ----- | ----- | ----- | ----- | ----- | ----- | ----- | ----- |
| Wind speed | m s$^{-1}$ | S-WSB-M003 (Onset, USA) | 5 | ----- | ----- | ----- | ----- | ----- | ----- | ----- | ----- | ----- | ----- | ----- | ----- | ----- | ----- | ----- | ----- | ----- | ----- |
| Wind direction | ° | S-WDA-M003 (Onset, USA) | 5 | ----- | ----- | ----- | ----- | ----- | ----- | ----- | ----- | ----- | ----- | ----- | ----- | ----- | ----- | ----- | ----- | ----- | ----- |
| Groundwater level**** | m.a.s.l. | DCL532 (BD sensors, Germany) | 15 | | | | | | | | ----- | ----- | ----- | ----- | ----- | ----- | ----- | ----- | ----- | ----- | ----- |
| Groundwater table depth**** | cm | DCL532 (BD sensors, Germany) | 15 | | | | | | | | ----- | ----- | ----- | ----- | ----- | ----- | ----- | ----- | ----- | ----- | ----- |
| Groundwater temperature**** | °C | Dallas DS 18B20 | 15 | | | | | | | | ----- | ----- | ----- | ----- | ----- | ----- | ----- | ----- | ----- | ----- | ----- |

*Net soil/ditch fluxes for all collars 1 - 27.
**Time in between two consecutive flux measurements. The 10 minutes comprise actual flux measurement of 5 minutes and 5 minutes headspace flushing between flux measurements.
***Measured for a subset of collars: 4, 7, 9, 23, 27.
****Measured for a subset of collars: 1, 5, 10 (ditch), 13, 18, 22, 27.

**2.3 The SkyLine2D system at Vejrumbro**

The SkyLine2D system is an automated chamber based system for measuring GHG fluxes. The system is designed and built by Earthbound Scientific Ltd. (United Kingdom). We used the SkyLine2D system to measure the net soil fluxes of $CO_2$, $CH_4$ and $N_2O$ measured with an automated GHG chamber system over 12 months resolving spatiotemporal patterns of GHG fluxes including 27 individual collars (26 on organic soil and 1 in a ditch) over a 24 m transect on a temperate drained fen peatland (Fig. 1B and 3).

The SkyLine2D system transect was oriented in a north-south direction (Fig. 1B). Two 2.5-meter-tall scaffold towers marked the end of the 30 m SkyLine2D system (Fig. 1B and Fig. S2D). The towers were fixed by ropes attached to 1000L pallet tanks filled with water (Fig. S2D) that maintained a stable position of the towers and ropes and hence placement of the chamber over the collars. The GHG analyser (model G2508, Picarro Inc., USA) was installed in a waterproof and temperature-controlled shelter at the south end of the transect (Fig. 1B and Fig. S2C). The transect was situated on the edge of the riparian fen near the mineral upland soils, where active agriculture was practiced (Fig. 1B). Along the transect volumetric soil water content (SWC) and soil temperature (ST) as well as water table depth (WTD) were measured at seven locations (Fig. 2). The agricultural field north of the SkyLine2D was sown with annual crops in rotation according to common practice.

2.3.1 Greenhouse gas flux measurements with the SkyLine2D system

Along the SkyLine2D transect the 26 individual collars (Ø19 cm) along the 24 meters transect on organic soil (Fig. 2) were inserted 5 cm into the peat leaving 5 cm above the surface. The collars were distanced app. 70 cm apart. One collar was installed in the ditch by inserting a tube (Ø19 cm, length 100 cm) to the bottom of the ditch with holes deeper than the minimum water level in the ditch to allow water flow. Thus, it was avoided that air entered in the collar in the ditch due to low water levels in the ditch. On top of this longer tube a collar (Ø19 cm, length 10 cm) was glued allowing for flux measurements. The chamber was programmed to stop when the bottom of the chamber sat the water surface if the water level in the ditch extended above the top of the collar. For most of the time the collar was not submerged, and the chamber therefore hit the collar.

There was one round transparent chamber (height: 39.5 cm and inner Ø: 19 cm, volume: 11.2 L) on the SkyLine2D, hanging below a moving trolley, which was suspended on two ropes stretched between the north and south towers (Fig. S2A and B). At defined positions along the rope, neodymium magnets had been inserted, and a magnet sensor (Fig. S2B) on the trolley informed the internal computer to stop and lower the chamber over positions with a collar on the surface. The chamber was lowered and guided down to the collar by supporting rods shaping a funnel (Fig. S2A). The chamber stopped when it hit the collar, achieved through a pressure sensor on top of the chamber connected to a hollow rubber gasket (Ø 3 cm) at the bottom, which also sealed the chamber with the collar. There was no fan installed in the chamber as the mixing was ensured by the main pump (Fig. S2C). A vent was installed in the top of the chamber to allow for pressure equilibration under windy conditions and chamber deployment.

One entire flux + flushing sequence lasted 10 minutes (Table 1). The chamber closure period was set to 5 minutes with a purging time of 5 minutes in between measurements when chamber was open and hanging underneath the trolley at approximately 1 meter above the ground (Fig. S2D). This provided on average 10 min

between flux measurements on consecutive collars (Table 1). Due to small variations in mechanical operations,
flux measurements were occasionally farther apart than 10 minutes, but overall, the timing of the SkyLine2D
system was consistent. After each cycle of 27 flux measurements there was a 30-minute delay until the start of
the next cycle. On average this resulted in 4-5 flux measurements per collar per day throughout the period.
To determine the concentrations of $CO_2$, $CH_4$ and $N_2O$ in the chamber air, a laser spectroscopy GHG analyser
(model G2508, Picarro Inc., USA) was used. The sample output frequency was set to 1 Hz with a manufactured
specified raw precision on 1 Hz data for $CO_2$: 240 ppb, $CH_4$: 0.3 ppb and $N_2O$: 5 ppb at ambient conditions
(Picarro Inc., USA). A main pump (model: N86 KN.18, KNF, Germany) circulated the air to and from the
chamber at 6 L $min^{-1}$. The GHG analyser was installed in parallel to the inflow from the chamber due to the
much lower flow of 250 mL $min^{-1}$ of the vacuum pump. There was a 30-meter tube between the chamber and
main pump to allow for the GHG analyser to remain stationary in the hut while the trolley moved.

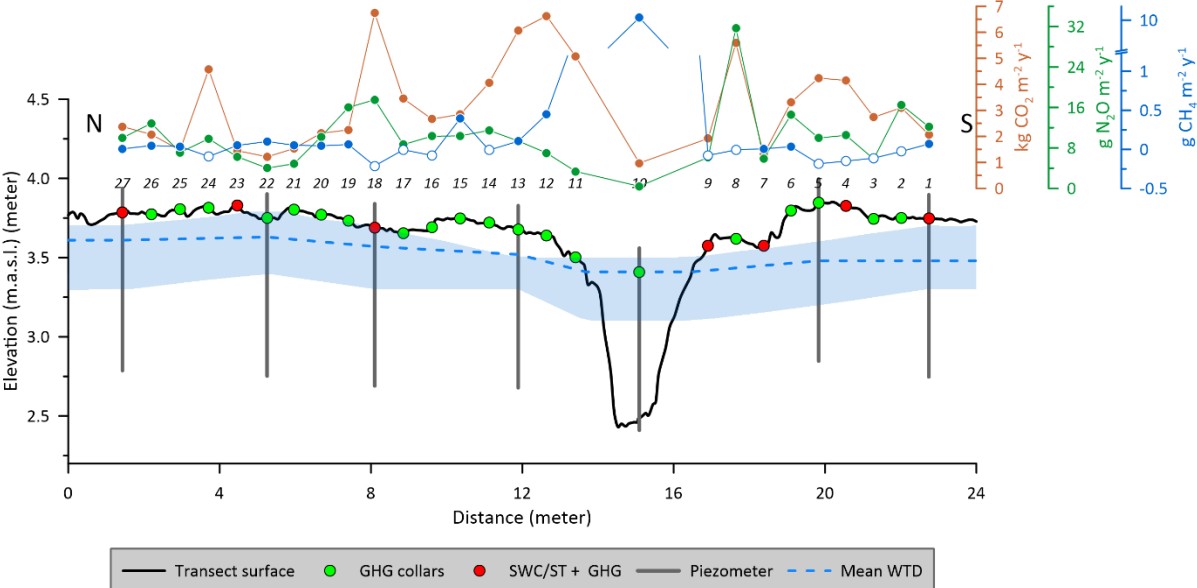


*Figure 2: Schematic* **representation of the measurement transect at Vejrumbro and associated measurement**
**variables. The annual cumulative fluxes of CO₂ (red) (kg CO₂ m⁻² y⁻¹), N₂O (green) (g N₂O m⁻² y⁻¹) and CH₄ (blue) (g**
**CH₄ m⁻² y⁻¹) are shown for each collar across the measurement transect at Vejrumbro. Closed and open symbols for**
**CH₄ represent net cumulative emission and uptake, respectively. Mean groundwater level is the mean water table**
**elevation measured in piezometers (blue dashed line) with shaded blue area represent maximum and minimum**
**observed groundwater elevation. GHG collars (green symbols) mark the positions of greenhouse gas flux**
**measurements of CO₂, CH₄ and N₂O. SWC/ST + GHG mark the positions where volumetric soil water content**
**(SWC) and soil temperature (ST) at 5 cm depth were measured alongside greenhouse gas fluxes. Numbers on top of**
**plot show the collar numbers (from 1 – 27). N and S mark the north and south ends of the transect (see Fig. 1B). The**
**peat depth was at least one meter in all points. Elevation is given meters above sea level (m.a.s.l.).**

**2.4 Peat sampling and analysis**

In November 2023 the peat across the SkyLine2D transect was sampled to 1 meter depth using a Russian auger and cores split into five layers of 20 cm thickness. Collars 1, 2, 5, 6, 8, 13 – 27 were sampled (Fig. 2). For the remaining collars it was not possible to retrieve a sample due to excessive wetness of the peat. The decomposition of the peat samples were assessed by a 10-point Von Post scale of humification (1 = completely undecomposed and 10 = completely decomposed) together with $pH_{H_2O}$ determined by suspending peat in demineralized water (1:5 peat:water mix), dry bulk density (g cm$^{-3}$) and total C and N by dry combustion (g C/N 100 g peat$^{-1}$ or %).

**2.5 Groundwater table level, depth and sampling**

To measure the groundwater level piezometers (inner diameter 5 cm) were installed at collars 1, 5, 10 (ditch), 13, 18, 22, 27 (Fig. 2) to 1 meter depth below the surface, which is deeper than the lowest groundwater level in summer (~60 cm below the surface) with openings from 0.1 – 1.2 meter below terrain. In the ditch the piezometer bottom was deeper than one meter to secure anchoring in the peat. The piezometers were installed approximately 50-60 cm beside the collars to avoid interference with the SkyLine2D system. After installation, piezometers were cleaned and sealed at the surface with bentonite pellets to avoid surface infiltration along the piezometers which can distort water level measurements.

Pressure transducers (Table 1) connected to Arduino-loggers were installed in each piezometer (at collars 1, 5, 10, 13, 18, 22 and 27 – Fig. 2) approximately 1 m below terrain measuring water levels every 15 minutes. The pressure transducers were vented and thus do not need correction for atmospheric pressure.

The groundwater levels were described using two metrics: hydraulic head and groundwater table depth (WTD). Hydraulic head represents the water level relative to mean sea level, based on the Danish Vertical Reference (DVR90), while WTD indicates the depth of the groundwater below the surface terrain and represented in positive values, where WTD of zero is equivalent to groundwater level at the terrain surface. The elevation of top of the piezometers were measured using a GPS (model GS07 High Precision GNSS Antenna with a CS20 Controller, Leica, Germany) and used as a local reference for hydraulic head. Manual measurements of groundwater levels were conducted every 2 months and used to calibrate the logger water levels to hydraulic head and WTD.

2.5.1 Groundwater water sampling and chemical analysis

Groundwater was sampled monthly in the piezometers placed at collars 1, 5, 13, 18, 22 and 27 (Fig. 2) by retrieving a 200 mL sample 20-30 cm below the groundwater level at the sampling time. The water sample was retrieved using a syringe and transferred to a plastic bottle that was capped immediately to avoid echange with the atmosphere and contamination. Water samples were frozen immediately after sampling and subsequently after thawing analyzed for pH, EC and alkalinity on an 855 Robotic Titrosampler (Metrohm, Germany). Total N and DOC were measured on a TOC-V CPH Analyzer with Total Nitrogen Unit TNM-1 & ASI-V Autosampler (Shimadzu, Japan). Ion chromatograph (IC) analyses of Cl$^-$, NO$_3^-$, and SO$_4^{2-}$ were performed on a 930Compact IC Flex (Metrohm, Germany) and NH$_4^+$ concentrations were measured with continuous flow analysis using a Seal AA500 Autoanalyzer (SEAL Analytic, USA). Total dissolved Fe and P were analyzed with coupled

plasma–mass spectrometry (ICP-MS) on an iCAP-Q ICP-MS (Thermo Fisher Scientific, USA) in KED mode

using He as the collision gas. Prior to analysis the 10 mL subsamples were acidified with 200 µL concentrated

nitric acid. Elemental ICP-MS analyses also included dissolved base cations of $Ca^{2+}$, $Mg^{2+}$, $K^+$, $Na^+$ as well as

total dissolved Al and Mn cations (not shown but included in the data set).

### 2.6 Soil moisture and temperature measurements

Soil moisture and temperature probes were initially inserted for collars 1, 4, 7, 9, 18, 23, 27 (Fig. 2) in order to

obtain a representation of the entire transect. Soil moisture probes (6 cm length) were inserted at an approximate

30˚ angle 5 cm outside the collar, while the soil temperature probes were inserted vertically adjacent to the soil

moisture probe. Due to sensor failures soil moisture was measured for collars 1, 7, 9, 18, 23 and 27 and soil

temperature at 4, 7, 9, 23 and 27.

### 2.7 Wireless data transfer

Wireless sensors for air temperature, wind speed, wind direction, soil temperature and volumetric soil water

content were set up with Wi-Fi data transfer to HOBO RX3000 Weather Station (Onset, USA) equipped with

HOBOnet Manager (RXMOD-RXW-868) module for wireless communication with sensors and logged data

every 5 minutes. Data access was through the HOBOlink cloud software.

Groundwater loggers were interfaced with the $I^2C$ (Inter-integrated Circuit) protocol and data was collected on

Arduino custom-built logger (https://vandstande.dk/logger.php) with wireless connection via LoRaWANor

SigFox.

### 2.8 Calculation of diffusive fluxes

Fluxes were calculated and quality checked using the goFlux R package (Rheault et al., 2024) and presented as

µmol $CO_2$ $m^{-2}$ $s^{-1}$, nmol $N_2O$ $m^{-2}$ $s^{-1}$ and nmol $CH_4$ $m^{-2}$ $s^{-1}$. Prior to flux calculations, the gas concentration data

from the G2508 analyzer was matched to the chamber closure time and chamber id to determine the start time of

the chamber measurement, so it was possible to separate individual flux measurements from each collar over the

measurement time (see examples of flux detection and calculation in Fig. S3A-D). An automatic deadband

detection method was applied based on maximal $R^2$ of a linear regression over the first 180 s (in 10 s steps) after

chamber closure. The deadband was allowed to attain values between 0 to 150 seconds thereby also allowing for

compensation for the ~60 s delay between chamber headspace gas concentration change and GHG analyser

detection due to transport time through the 30 m tube connecting the chamber and GHG analyser.

Flux calculations were done with both linear (LM) and non-linear (Hutchinson-Mosier – HM) regression models

(Pihlatie et al., 2013) to determine the slope at time zero. The best flux estimates with either the LM or HM

regression model was determined using the *best.flux* function in the goFlux package (Rheault et al., 2024).

Shortly, if the RMSE of the HM model was lower than minimum detectable flux (MDF), HM was chosen.

However, if the ratio (g-factor) between HM and LM was larger than 2, LM was chosen, as this indicates over-

fitting of the HM, which may result in unrealistic large HM flux estimates. If the relative SE of the slope

(SE/slope) at time zero for the HM model was larger than 100% it indicated overfitting of the HM model and

the LM was chosen. This approach is conservative as it will discard non-linear flux behaviour and instead

provide a conservative linear flux estimate. Out of 47.438 detected flux measurements for $CO_2$, $CH_4$ and $N_2O$,

respectively, a total of 2807 $CO_2$ fluxes (5.9%), 3339 $N_2O$ fluxes (7%) and 4923 $CH_4$ fluxes (10.3%) were
discarded due to the following two situations: 1) chamber mechanical malfunction either resulting in imperfect
sealing on collar due to erroneous lowering of chamber on collar indicated by background atmospheric or
fluctuating gas concentrations in the headspace and 2) at *in situ* flux levels close to the minimum detectable flux
of the Picarro G2508 analyser (Christiansen et al., 2015) non-significant regression (between concentration and
time and GHG concentration) ($p>0.05$) were also discarded as it was not possible to statistically distinguish
whether there was a real flux or the lack of significant regression was because of chamber malfunction. It is
acknowledged that discarding low fluxes can bias annual means and cumulative values, but the data quality did
not allow us to determine whether the flux measurement was performed correctly and hence a conservative
approach was chosen as including false low fluxes would also bias the data set.
For flux measurements the air temperature in 2 meters was used as an estimate of the chamber headspace
temperature along with a 1 atm air pressure.
The annual cumulated fluxes from the soil or the ditch (diffusive only) were estimated simply by multiplying the
daily average $CO_2$, $CH_4$ or $N_2O$ flux for the measurement period with 365 days. We believe for the purpose of
data presentation that this simplistic methodology is adequate here, also given the very few data gaps in the
timeseries. However, there are other more sophisticated methods using interpolation and response variable
functions that may refine the annual budget. However, it is not the goal of this manuscript to present these
methodologies but to provide the data so other users can test different temporal upscaling methodologies.

### 2.9 Calculation of ebullition fluxes in the ditch

Methane ebullition fluxes were occasionally observed only in the ditch. The resultant $CH_4$ time series for the
chamber would have a characteristic appearance (Fig. S4) where the measurement would essentially start out as
diffusive flux measurement, then $CH_4$ bubbles entered the chamber headspace, and the concentration would
quickly increase to a maximum value and reach a threshold concentration corresponding to the mixed headspace
concentration. In these cases, the LM/HM flux calculation assumptions are violated and instead the ebullition
flux would be calculated as the total increase in $CH_4$ mass $m^{-2}$ per 5 min enclosure. The mass flux of $CH_4$ per
enclosure (nmol $m^{-2}$ per 5 min enclosure) was calculated according to Eq. (1):

$$F_{CH_4-ebu} = dCH_4 * \frac{V_{system}*P}{A*R*T} \text{ (1)}$$

Where $dCH_4$ is the concentration difference in nmol between start of chamber enclosure ($CH_{4,start}$) and end $CH_4$
concentration ($CH_{4,end}$) after it reached a plateau (Fig. S4), $V_{system}$ is the total volume (11.7 L) of the system
(collar, chamber, tubes and GHG analyser) in L, P is the pressure (1 atm), A is the area of the collar (0.028 $m^2$),
R is the gas constant (0.082057 L atm $K^{-1}$ $mol^{-1}$) and T is the chamber headspace temperature (K). To calculate
the ebullition flux per second the ebullition flux estimate was divided by 12*60 seconds (300), equivalent to the
number of seconds over the 5 minute measurement period.
Out of a total of 1728 flux measurements from the ditch (collar 10), 334 were classified as ebullitions according
to our definition above. indicating that ebullition was erratic which is in line with studies of ebullition of fluxes
from ponds (Sø et al., 2023; Wik et al., 2016). Hence, it can be assumed that ebullition occurred around 19.3%
of the time during the measurement period (360 days). An annual estimate of the ebullition flux was calculated
as the average ebullition flux in nmol $CH_4$ m$^{-2}$ s$^{-1}$ by multiplying with number of seconds over 365 days and the
19.3% during period where ebulittion occurred.
Ebullitions could also be caused by mechanical disturbance of the chamber landing on the collar. Ebullition
fluxes were discarded if the sudden increase in $CH_4$ headspace concentration (Fig. S4) occurred 60 seconds after
recorded chamber closure as this indicated bubbles released by chamber deployment on top of the collar.
**3 Data presentation**
**3.1 Wind speed and direction**
Generally, the wind regime during the measurement period (February 2$^{nd}$, 2022 to January 28$^{th}$, 2023) was rather
mild with monthly average wind speeds ranging between 1.2 to 2.9 m s$^{-1}$ and maximum gust up to 20 m s$^{-1}$. The
wind direction was uniformly from the west for 52% of the time, with easterly winds constituting 27% and
northern and southern winds 8 and 13% of the time (Fig. 3). Winds from western directions were highest for the
longest period, while easterly winds were of similar magnitude, but less frequent (Fig. 3). Northern and
southerly winds were generally below 3 m s$^{-1}$ and represented periods with still conditions. The very uniform
western-eastern wind field at Vejrumbro may also partly be explained by the W-E direction of the valley in
which the site is situated, that effectively blocks or dampens winds from S and N.

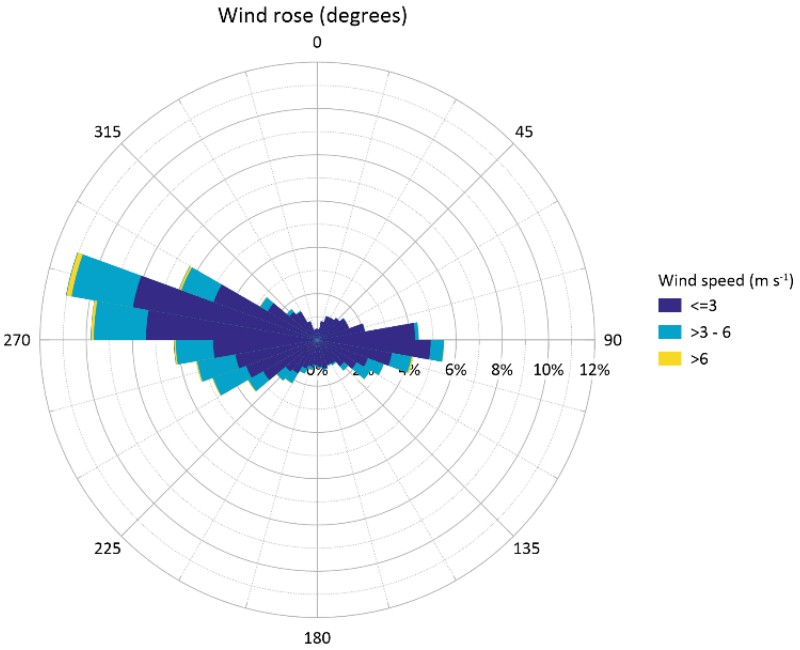


**Figure 3: Wind regime at Vejrumbro for the period July 1$^{st}$, 2021 to January 31$^{st}$, 2023 presented as a wind rose**
**diagram with wind speed and direction for the period.**

**3.2 Air and soil temperature**

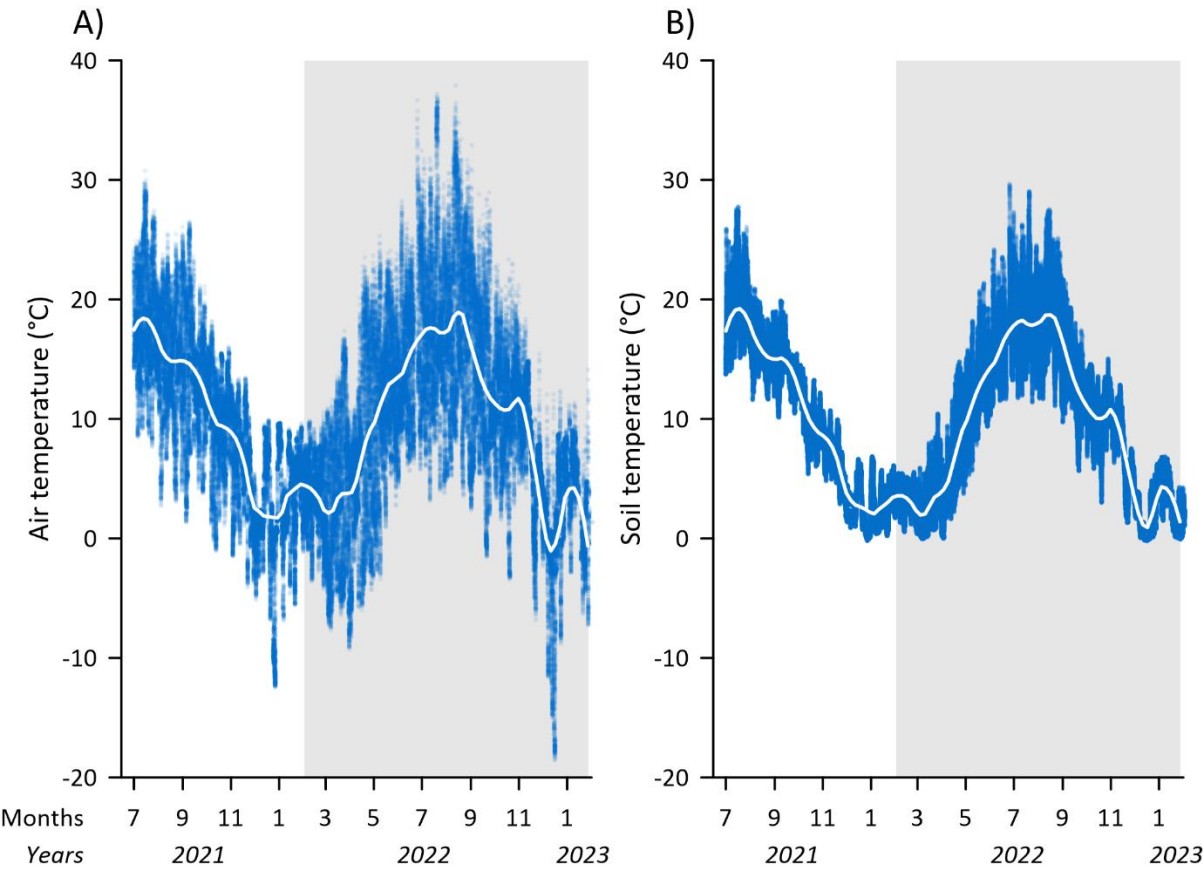

**Figure 4: Time series of A) air temperature in °C measured at 2 meter height above the surface and B) soil temperature (°C) at 5 cm depth for collars 4, 7, 9, 23 and 27 along the measurement transect. The blue dots are the raw 5 min measurements of air temperature and the white lines represent LOESS fit to show overall seasonal trend. The periods of GHG measurements with the SkyLine2D system are shown with the shaded area.**

Over the study period the average air temperature was 9.6°C ranging between maximum 37.9°C and minimum of -18.6°C (Fig. 4A). Monthly ranges of air temperatures (Table 2) show >20°C variation between minimum and maximum, except for February, pointing towards large diurnal variations. Soil temperature magnitude and temporal variation were similar across the transect, varying between 0 to 28°C (Fig. 4B) and followed that of air temperature (Fig. 4A) with less variability (Fig. 4B and Table 2). The annual site average soil temperature was similar to the air temperature (Table 2).

**Table 2: Monthly mean, maximum and minimum air temperature and soil temperature (°C), groundwater table**
**depth (cm) and volumetric soil water content (cm$^3$ cm$^{-3}$) at Vejrumbro in the measurement period from February 1$^{st}$,**
**2022 to January 31$^{st}$, 2023.**

| Variable | Year Month | 2022 Feb | Mar | Apr | May | Jun | Jul | Aug | Sep | Oct | Nov | Dec | 2023 Jan | Avg |
|---|---|---|---|---|---|---|---|---|---|---|---|---|---|---|
| **Air temperature (°C)** | **Mean** | 3.8 | 3.0 | 6.6 | 12.0 | 15.4 | 17.7 | 16.6 | 13.4 | 10.7 | 6.9 | 1.2 | 3.7 | 9.6 |
| | **Max** | 10.6 | 17.4 | 23.7 | 25.3 | 36.7 | 37.2 | 37.9 | 32.9 | 23.3 | 18.4 | 12.4 | 14.1 | - |
| | **Min** | -4.3 | -9.3 | -8.3 | -3.4 | 4.3 | 3.2 | 2.7 | -1.5 | -3.5 | -6.9 | -18.6 | -7.3 | - |
| **Soil temperature (°C)** | **Mean** | 3.0 | 3.2 | 2.9 | 6.4 | 12.3 | 16.1 | 18.4 | 17.0 | 13.8 | 10.3 | 7.2 | 2.1 | 9.6 |
| | **Max** | 6.5 | 5.3 | 9.1 | 12.5 | 18.8 | 25.1 | 27.0 | 24.7 | 19.3 | 14.3 | 12.6 | 6.3 | - |
| | **Min** | 0.3 | 1.1 | 0.4 | 0.8 | 6.6 | 10.7 | 12.4 | 11.8 | 7.0 | 4.0 | 2.1 | 0.0 | - |
| **Groundwater table depth (WTD) (cm)** | **Mean** | - | 39 | 35 | 41 | 36 | 41 | 35 | 31 | 20 | 18 | 17 | 13 | 29 |
| | **Max** | - | 58 | 39 | 58 | 43 | 52 | 46 | 36 | 30 | 31 | 28 | 28 | - |
| | **Min** | - | 23 | 5 | 24 | 9 | 28 | 22 | 9 | 5 | 6 | 3 | 2 | - |
| **Volumetric soil water content (cm$^3$ cm$^{-3}$)** | **Mean** | 0.53 | 0.45 | 0.40 | 0.37 | 0.38 | 0.43 | 0.43 | 0.45 | 0.50 | 0.53 | 0.52 | 0.51 | 0.46 |
| | **Max** | 0.56 | 0.51 | 0.50 | 0.41 | 0.47 | 0.55 | 0.56 | 0.56 | 0.57 | 0.58 | 0.56 | 0.57 | - |
| | **Min** | 0.43 | 0.39 | 0.37 | 0.33 | 0.32 | 0.26 | 0.32 | 0.35 | 0.40 | 0.47 | 0.42 | 0.34 | - |

**3.3 Groundwater table depth**
Average groundwater table depth (WTD) below terrain during the period was between 47 to 21 cm across the
transect (Fig. 2, Table 2). During summer, the peat drained between 18 – 31 cm below the annual average and in
winter the WTD increased to 0 – 22 cm above the annual average across the transect (Fig. 2, Table 2).
Generally, the WTD elevation was lower in the ditch across the entire study period (Fig. 2). It was only on the
northern end of the transect that the surface occasionally was flooded during winter periods (Fig. 2).

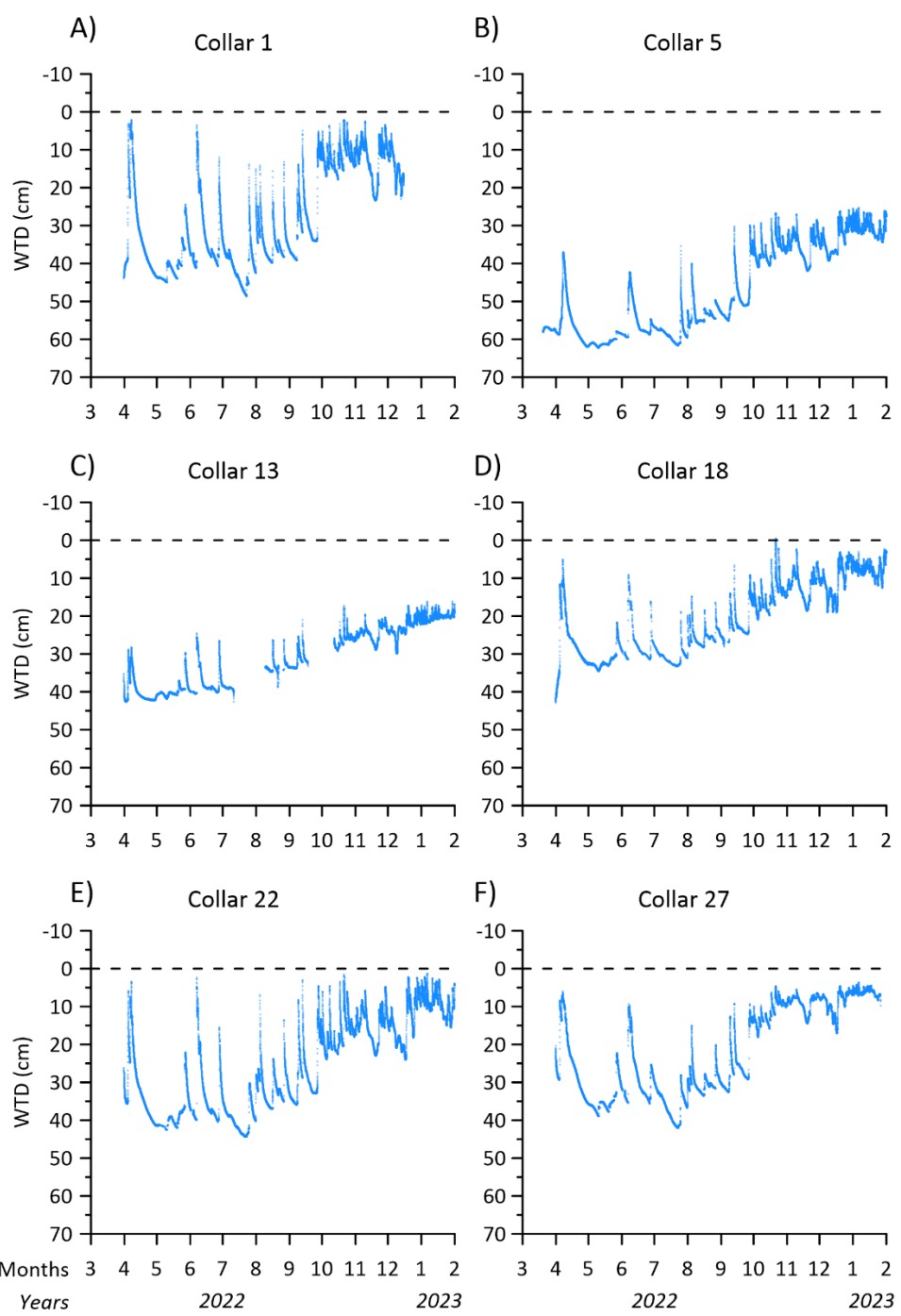

**Figure 5: Time series of groundwater table depth (WTD) below terrain for the six piezometer locations along the SkyLine2D transect in the period March 31st, 2022 and January 31st, 2023 when the flux measurements stopped. Dashed line show surface.**

The temporal variability of WTD was similar across the transect despite different absolute water table depths (Fig. 5A-F). In the summer periods, the WTD was most variable decreasing to below 40 cm for collars 1, 13, 18, 22 and 27, whereas the WTD for collar 5 showed the deepest WTD measured over the transect. WTD responded quickly (within hours) to precipitation events that could increase the WTD by almost 40 cm at some plots, indicating that the entire aerated soil volume above the groundwater table was flooded. There was a slight tendency to lower response to precipitation events for piezometers at collar 5 and collar 13 that were placed

closer to the ditch (Fig. 2 and Fig. 5B and C). As the ditch water level was lower than in the peat this could be explained by more efficient lateral drainage into the ditch from the areas closer to the ditch. In the winter periods, the WTD was less responsive to precipitation and was closer to the surface (Fig. 5A-F) across the transect.

**3.5 Soil water content**

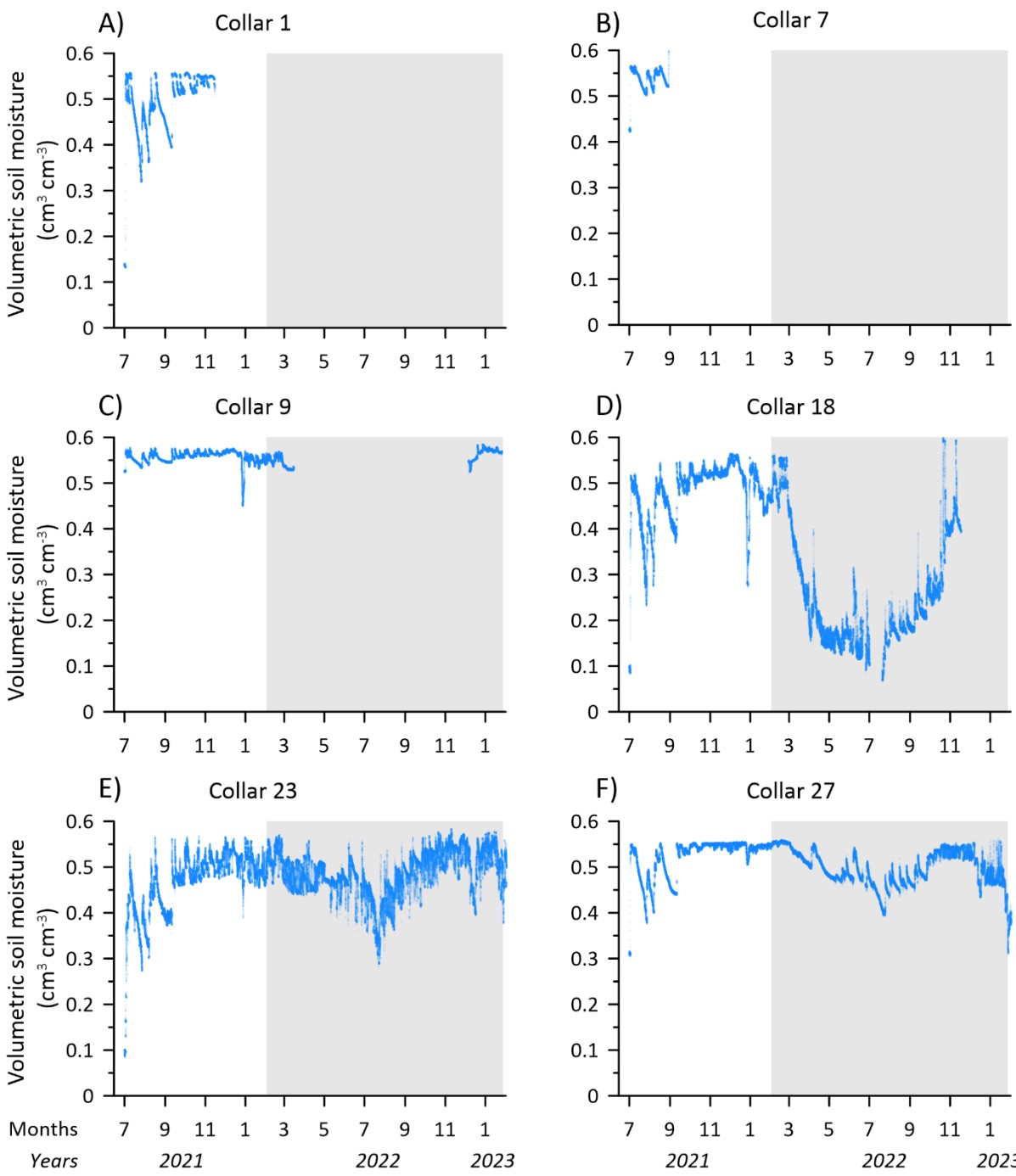

**Figure 6: Time series of volumetric soil water content (cm$^3$ cm$^{-3}$) in 0-5 cm for the six collars 1, 7, 9, 18, 23 and 27 along the SkyLine2D transect in the period July 1$^{st}$, 2021 – January 31$^{st}$, 2023 when the measurements terminated. The periods of GHG measurements with the SkyLine2D system are shown with the shaded area.**

Due to instrument failure the temporal coverage of soil moisture in the topsoil (5 cm) was not similar across the transect (Fig. 6A-F). For collars 18, 23 and 27 the entire period of greenhouse gas measurements was covered by soil moisture measurements (Fig. 6D-F). While SWC for collars 1, 9, 18, 23 and 27 was similar in the winter periods (around 0.55 $cm^3$ $cm^{-3}$) the SWC for collar 18 decreased to lower minima between $0.1 - 0.2$ $cm^3$ $cm^{-3}$, than the minima observed between $0.3 - 0.4$ $cm^3$ $cm^{-3}$ for collars 23 and 27 in the summer periods (Fig. 6, Table 2). Similar for all collars it was observed that SWC was more variable in summer, responding similarly as WTD to precipitation events (Fig. 6, Table 2). Since plants were removed regularly from the collars the decrease of SWC for collar 18 cannot be explained by plant transpiration, and the dynamic behaviour could indicate the impact of soil evaporation, but the different levels of SWC also show that there is spatial variation across the transect in the water retention properties of the peat soil that will impact the rate of drying. However, it cannot be ruled out that the SWC sensor at collar 18 experienced malfunction or that soil contact was lost in the dry periods of 2022 (Fig. 6D) which could lead to erroneous and too low SWC. Therefore, these data should be considered with care.

### 3.6 Peat soil characteristics

**Table 3 Mean (±standard error of the mean (SE)) peat/organic soil characteristics of humification degree (Von Post), pH ($H_2O$), dry bulk density ($\rho_{dry}$), total C (TC) concentration, total N concentration (TN) and the C/N ratio for collars 1, 2, 5, 6, 8 and 13 - 27 at the Vejrumbro transect.**

| Depth (cm) | N | Von post | | pH ($H_2O$) | | $\rho_{dry}$ (g $cm^{-3}$) | | TC (%) | | TN (%) | | C/N | |
|---|---|---|---|---|---|---|---|---|---|---|---|---|---|
| | | Min | Max | Mean | ±SE | Mean | ±SE | Mean | ±SE | Mean | ±SE | Mean | ±SE |
| 0-20 | 20 | 7 | 10 | 4.2 | 0.08 | 0.31 | 0.02 | 26 | 1.1 | 1.6 | 0.06 | 16 | 0.4 |
| 20-40 | 20 | 5 | 10 | 4.6 | 0.06 | 0.20 | 0.01 | 43 | 1.3 | 1.8 | 0.04 | 24 | 0.7 |
| 40-60 | 11 | 3 | 8 | 4.9 | 0.10 | 0.15 | 0.01 | 48 | 1.8 | 1.9 | 0.05 | 25 | 1.1 |
| 60-80 | 11 | 3 | 6 | 5.3 | 0.09 | 0.11 | 0.01 | 47 | 1.8 | 1.9 | 0.05 | 24 | 0.6 |
| 80-100 | 10 | 1 | 8 | 5.4 | 0.09 | 0.10 | 0.02 | 44 | 2.1 | 1.9 | 0.05 | 24 | 0.6 |

Generally, there was peat/organic soil to one meter depth except for one collar (25) where gyttja was found in a depth of 80 cm (Table 3). The organic soil was more decomposed in the top 40 cm indicated by higher Von Post values between 5 and 10. Below 40 cm peat still displayed high levels of decomposition along the transect, but was more often found to be less decomposed, values ranging from 1-8 (Table 3). This corresponds well to the previous land use with drainage of the topsoil leading to higher degree of humification. Also, the organic soil was most dense in the top 20 cm (on average $0.31\pm0.02$ g $cm^{-3}$) and bulk density decreased to $0.10 - 0.12$ g $cm^{-3}$ from $40 - 100$ cm depth. Total C and N was lowest in the 0-20 cm layer, but still classified as organic soil. Below 20 cm total C and N concentrations, respectively were similar. C/N ratio was lowest in the top 20 cm ($16\pm0.4$) and increased to 22-25 in $20 - 100$ cm depth (Table 3).

### 3.7 Groundwater and ditch water chemical composition

Site mean pH of the groundwater in the organic soil was $5.8\pm0.1$ and was lower than the pH of the ditch ($7.3\pm0.6$). There was a tendency towards lower pH in groundwater and ditch towards the end of the measurement period (Fig. 7A). Electric conductivity was generally higher in the ditch water ($359\pm36$ µS cm-1) compared to the groundwater in the organic soil ($276\pm18$ µS $cm^{-1}$) but varied less over the season. The groundwater shows a clear peak in EC around September 2022 (Fig. 7B). Total dissolved P was markedly

higher in the groundwater (687±45 µg P L$^{-1}$) compared to the ditch water (76±10 µg P L$^{-1}$). Whereas there was
little seasonal trend in ditch P concentrations, dissolved P in groundwater dipped to below average
concentrations between August to October, likely indicating plant uptake during the growing season (Fig. 7C).
Similarly, total dissolved N was higher in groundwater (6.7±0.5 mg N L$^{-1}$) than in ditch (2.6±1.6 mg N L$^{-1}$) with
increasing concentrations during the growing season (Fig. 7D). This temporal trend was also observed for $NO_3^-$
(Fig. 7E), but average groundwater (2±0.5 mg $NO_3$-N L$^{-1}$) and ditch (2.2±1.5 mg $NO_3$-N L$^{-1}$) concentrations
were similar. As expected, dissolved NH4-N was lowest among investigated N-species and there was more
dissolved $NH_4$-N present in groundwater (0.8±0.1 mg $NH_4$-N L$^{-1}$) than in the ditch (0.14±0.25 mg $NH_4$-N L$^{-1}$).
However, there was no discernable temporal trend for $NH_4^+$ (Fig. 7F). Collectively, the temporal trend of TN
and $NO_3^-$ could point to temperature driven mineralization of the peat. Also, the organic N (TN – inorganic N-
species) was on average 10 times higher in the groundwater than in the ditch. Average $SO_4^{2-}$ concentrations
were similar between the groundwater (17.5±2.4 mg $SO_4$-S L-1) and ditch (17±1.5 mg $SO_4$-S L$^{-1}$), but $SO_4^{2-}$
concentration peaked during September and October in the groundwater whereas it remained more constant in
the ditch over the season (Fig. 7G). Like the dissolved organic N, DOC concentrations were consistently higher
in the groundwater (73±3.1 mg DOC L$^{-1}$) than in the ditch (9.4±3.5 mg DOC L$^{-1}$), but peaked later in the season,
around December 2022, whereas there was little temporal variability of DOC in the ditch (Fig. 7H). Dissolved
total Fe displayed the same temporal trend as DOC (Fig. 7I) but was higher groundwater (1916±163 µg Fe L$^{-1}$)
compared to the ditch (98±95 µg Fe L$^{-1}$). The geochemical parameters of groundwater and ditch water point to
different mechanisms regulating peat decomposition and possibly plant uptake, where the chemical composition
of groundwater varied more over time than ditch water. Generally, there were no systematic spatial pattern of
groundwater chemistry across the transect.

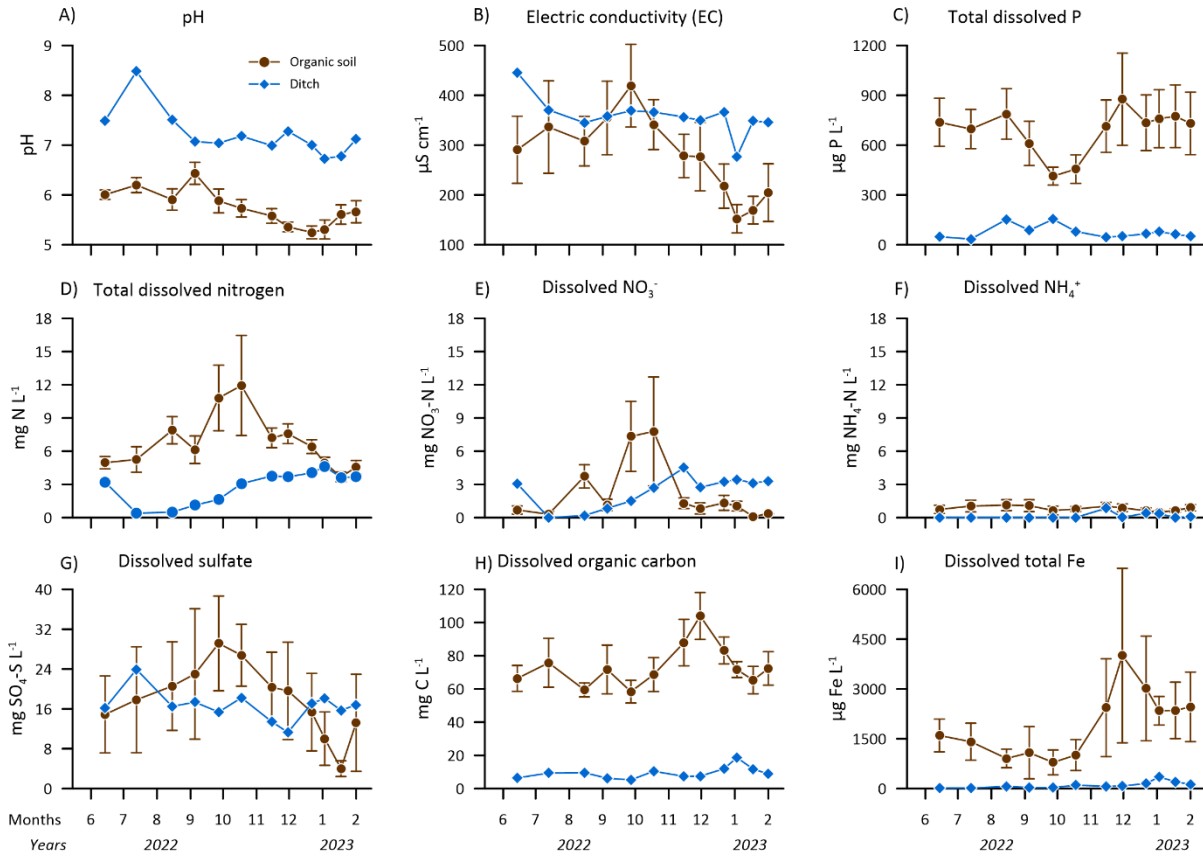


**Figure 7** Groundwater (brown closed circles) and ditch water (closed blue diamonds) chemistry at Vejrumbro for the period June 2022 to February 2023 for A) pH, B) Electric conductivity and dissolved C) total phosphor (P), D) total nitrogen (N), E) nitrate ($NO_3^-$), F) ammonium ($NH_4^+$), G) sulfate ($SO_4^{2-}$), H) organic carbon and I) total iron (Fe). Values for organic soils are means for the transect with error bars showing the standard error of the mean (N=6 per sampling date).

## 3.8 Net soil and ditch $CO_2$, $CH_4$ and $N_2O$ fluxes

### 3.8.1 Spatial variation of net soil $CO_2$, $CH_4$ and $N_2O$ fluxes

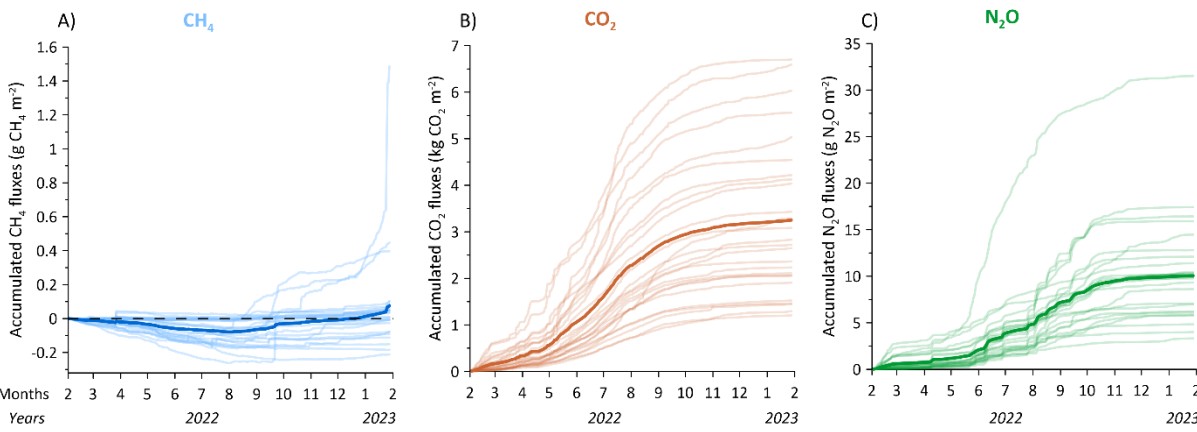

447

**Figure 8: Cumulative fluxes of A) $CH_4$, B) $CO_2$, and C) $N_2O$ for 26 individual collars along the SkyLine2D transect in the measurement period February 2022 to January 2023. Units for $CH_4$ and $N_2O$ are in g $CH_4$/$N_2O$ m$^{-2}$ and for $CO_2$ in kg $CO_2$ m$^{-2}$. The cumulative fluxes represent the raw dataset. The ditch data was excluded. Transect average is shown as thick lines.**

Within the transect, cumulative $CH_4$ fluxes over the study period (360 days) varied between -0.21 to 1.48 g $CH_4$
$m^{-2}$ over the study period, with a transect average (±SE) cumulative flux of 0.07±0.06 g $CH_4$ $m^{-2}$ (Fig. 2 and Fig.
8A). Out of the 26 collars, excluding the ditch collar, 11 displayed a net uptake over the measurement period
and the remaining were small net emitters (Fig. 2 and Fig. 8A). There was generally little spatial variation in the
absolute $CH_4$ fluxes among the soil collars, but three collars (11, 12 and 15) showed increasing net positive
cumulative fluxes towards the ditch (Fig. 2). The low spatial and similar temporal variation between collars
indicate both hydrological indicators of SWC and WTD are poor predictors of $CH_4$ fluxes across the transect.
However, as we excluded plants from the collars we might have decreased the net emission of $CH_4$ directly by
restricting gas transport in aerenchyma from deep peat layers potentially sustaining net $CH_4$ emission even
though the observed growing season WTD was 20-40 cm (Askaer et al., 2011; Vroom et al., 2022) and
indirectly by potentially reducing plant carbon supply to methanogens. The lack of consistent hot moments of
$CH_4$ emissions, low cumulative emissions during periods of shallow WTD in the growing season (Fig. 5A-F) is
in line with the measured free $NO_3^-$, $SO_4^-$, Fe ions (Fig. 7E, G, I) in the groundwater. It is well known that the
presence of other electron acceptors, such as sulphur, iron and nitrate, inhibit $CH_4$ production (Bridgham et al.,
2013), in turn limiting net $CH_4$ emission. Also, the often deeper WTD in the summer between to 40 cm below
terrain also suggest that $CH_4$ oxidation could aid to reduce net CH4 emission from the peat (Christiansen et al.,
468  2016).

The $CO_2$ effluxes displayed tremendous spatial variation across the 24-meter transect (Fig. 2 and Fig. 8B) and
measurements indicated that the organic soil was a net source of $CO_2$, with cumulative fluxes over the study
period  ranging between 1214 – 6740 g $CO_2$ $m^{-2}$, and a transect average (±SE) of 3269±328 g $CO_2$ $m^{-2}$, over the
study period of 360 days (Fig. 2 and Fig. 8B). There was no apparent relation between the magnitude of
cumulative $CO_2$ efflux to the position along the transect and average WTD (Fig. 2). The cumulative net soil $CO_2$
emission is equal to 8.9 t$CO_2$-C ha$^{-1}$ y$^{-1}$ (range of 3.3 to 18 t$CO_2$-C ha$^{-1}$ y$^{-1}$ across the transect) and compares
well to estimates of annual soil C loss (8.8 t$CO_2$-C ha$^{-1}$ y$^{-1}$) from a drained unfertilized grassland on organic soil
in Denmark (Kandel et al., 2018) as well as annual carbon budgets of similar Danish, British and German
wetlands (Evans et al., 2021; Koch et al., 2023; Tiemeyer et al., 2020).
Similarly, the partcular site at Vejrumbro where the SkyLine2D was located was overall a net source of $N_2O$,
with cumulative fluxes ranging between 3.3 – 32 g $N_2O$ $m^{-2}$, with a transect average (±SE) of 10.1±1.1 g $N_2O$ $m^{-}$
$^2$ (Fig. 2 and Fig. 8C) over the study period (360 days). Thus, there is a 10-fold difference between minimum
and maximum cumulative $N_2O$ fluxes within the transect, without any apparent relation to the position along the
transect and WTD. The highest cumulative $N_2O$ fluxes occurred at collar 8 situated close to the ditch (Fig.
2).The transect average cumulative $N_2O$ emission is equivalent to a net N loss from $N_2O$ emission alone of 64
kg N ha$^{-1}$ y$^{-1}$, was very high and exceeding previously reported fluxes from the Vejrumbro site (1.5 – 2.1 g $N_2O$
$m^{-2}$ y$^{-1}$) (Nielsen et al., 2024) and German organic soils (0.04 – 6.3 g $N_2O$ $m^{-2}$ y$^{-1}$ for grassland and cropland
land uses) (Tiemeyer et al., 2020). The high $N_2O$ emission from the transect during the measurement period
indicate that $N_2O$ may in fact dominate the GHG budget in relation to the global warming potential at this
specific location at the Vejrumbro site had gross primary production (reducing net ecosystem $CO_2$ emission)
been included in the measurements. It is important to reiterate here that the flux measurements of this study
were done on bare soil whereas the studies referenced above included vegetation.
The high $N_2O$ fluxes may be a result of high rates of denitrification in the subsoil from either *in situ* produced
$NO_3^-$ from peat decomposition or as $NO_3$-enriched agricultural runoff from the surrounding intensively
cultivated areas, which was not affecting groundwater $NO_3^-$ concentration in the center of the wetland with
lower $N_2O$ (Nielsen et al., 2024). The groundwater enters the northern peripheral zone of the wetland at
Vejrumbro coinciding with the position of the measurement transect. The highest $NO_3^-$ concentrations in
groundwater at the SkyLine2D transect corresponded roughly with highest $N_2O$ emissions during summer and
early autumn (Fig. 7D-F and Fig. 8C), but the frequency of water sampling was too low to fully link
groundwater $NO_3^-$ temporal dynamics to $N_2O$ emissions.

**499**   **3.8.2 Temporal variability of net soil CO₂, CH₄ and N₂O fluxes**

**500**   **3.8.2.1 Time series of raw data of net soil CO₂, CH₄ and N₂O fluxes**

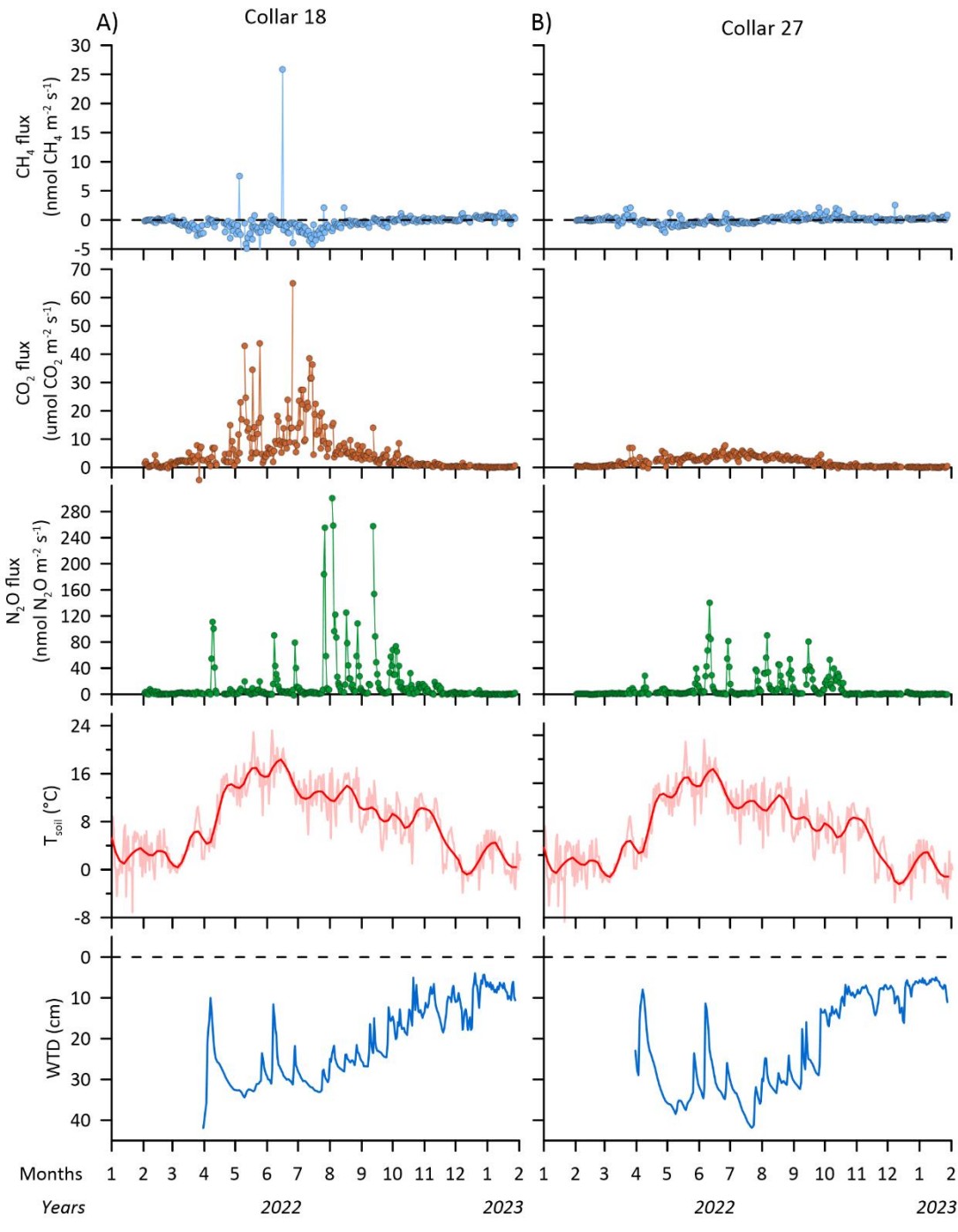

**502**   **Figure 9: Examples of daily average time series of CH₄, CO₂ and N₂O fluxes for collars 18 and 27 at the SkyLine2D**

**503**   **transect in Vejrumbro, soil temperature (Tsoil) in celsius (°C) and groundwater table depth (WTD) in cm below**

**504**   **terrain is shown in two lower panels for the measurement period from February 2022 to January 2023.**

**505**   With the high frequency of GHG flux measurements (on average 5 measurements per day per collar) it was

**506**   possible to observe short term flux phenomena that in most studies deploying manual chambers are missed or if

**507**   captured can lead to biased conclusions on flux magnitudes. For example, in most of the measurement points,

CH$_4$ fluxes were generally near zero, but occasionally displayed elevated net emission for short periods even in
periods with deeper WTD (Fig. 9A) for most chambers (see supplementary Fig. S5). This flux dynamic might
be related to episodic release of accumulated CH$_4$ from deeper soil layers that are not fully oxidized in the
aerated root zone and that were not released through plants (Askaer et al., 2011). As plants were not included in
the collars these bursts cannot be attributed to plant emission pathways.
Generally, it was observed that soil CO$_2$ fluxes increased over the season with increasing temperature. However,
for some collars displayed rapid bursts of CO$_2$ emissions (example in Fig. 9A), while other collars at the same
period did not display this behaviour (Fig. 9B). This dynamic points to different emission pathways from the
soil not related to plant mediated transport. Thus, while we purposely omitted aboveground autotrophic
respiration by clipping the vegetation, it cannot be ruled out that living roots inhabited the soil below the
chambers and hence contributed to the observed CO$_2$ emission rates.
For N$_2$O, the spatiotemporal pattern was even more pronounced than for CO2, with N$_2$O primarily emitted in
bursts related to rapidly increasing or decreasing WTD that coincided with precipitation events. In drier periods
with deeper WTD and little fluctuations, N$_2$O fluxes quickly dropped to near zero (Fig. 9A and B). Despite N$_2$O
being emitted in similar temporal patterns across the transect, the magnitude of the N$_2$O peaks were not similar
across the transect (Fig. 2, 8 and supplementary Fig. S5). Hence, the majority of N$_2$O is emitted in hot moments
is likely driven by fluctuations in WTD (Fig. 9) as it has also been shown in other drained temperate peatland
soils (Anthony and Silver, 2023).
**3.8.2.2 Diurnal variation of net soil CO$_2$, CH$_4$ and N$_2$O fluxes**

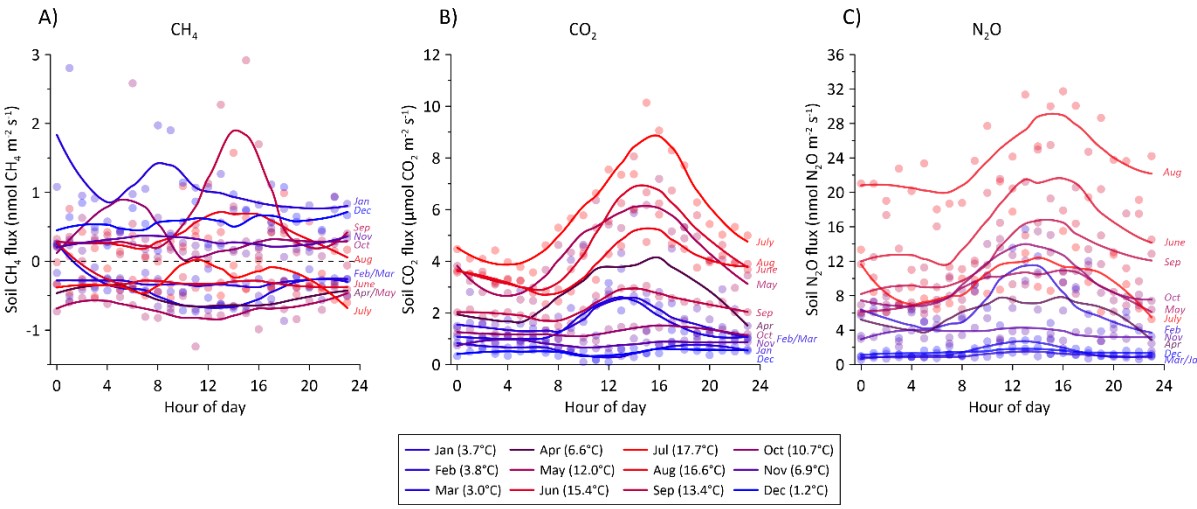


**Figure 10: Average hourly flux for all soil collars of A) CH$_4$, B) CO$_2$, and C) N$_2$O during a 24 hour period. The**
**diurnal variation is split between each month during the 2022-2023 measurement period. The fluxes were assigned**
**the hour of measurement during the day and averaged per month. Color shade between blue and red corresponds to**
**average air temperature for the specific month shown in parenthesis in the figure legend. Solid lines are loess fits for**
**visualization of the diurnal variation in each month.**
With the SkyLine2D system we observed a clear diurnal cycle for CO$_2$ and N$_2$O fluxes, but not for CH$_4$ (Fig.
10A-C). The lack of diurnal variability of CH$_4$ fluxes could also be due the removal of plants from the collars
that would have facilitated light-driven fluxes (Askaer et al. 2011). The amplitude of diurnal variability

increased with higher air temperature for $CO_2$ (Fig. 10B) and partly for $N_2O$ (Fig. 10C). The month of July was an exception as it resembled the pattern observed in May although the July soil temperature was about 5°C higher (Table 2). The lower $N_2O$ fluxes observed in July can be attributed to lower and more constant WTD in July compared to May, June and September across the transect (Fig. 5). Diurnal variability of soil $CO_2$ fluxes are well known and can be related to both increased heterotrophic respiration during the warmer day and autotrophic respiration in response to photosynthesis. Previously, similar diurnal patterns of $N_2O$ emissions were observed in a Danish fen (Jørgensen et al., 2012).

### 3.8.2.3 Monthly variability of net soil GHG fluxes

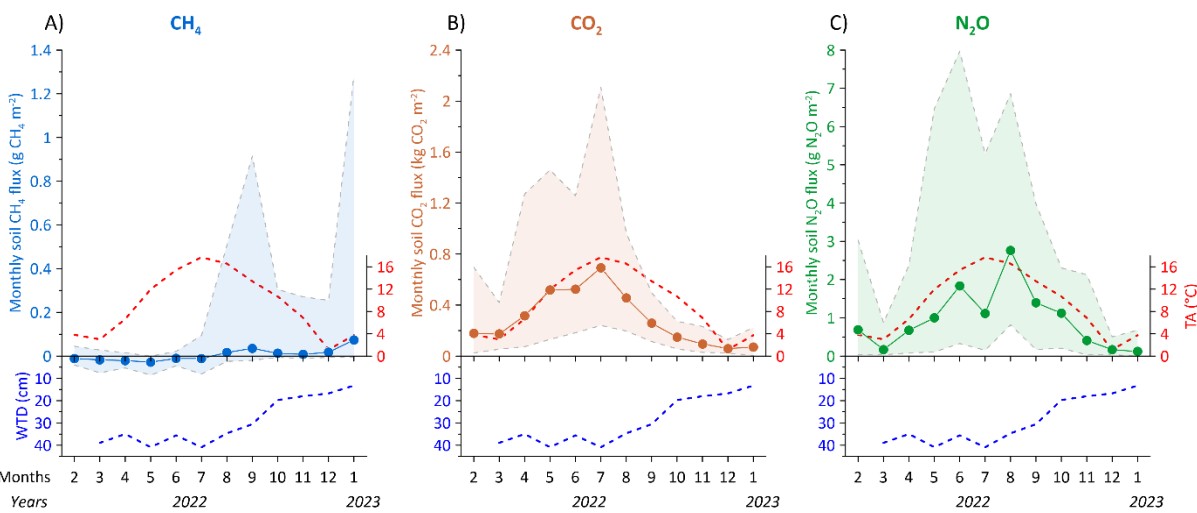

**Figure 11: Monthly summed soil fluxes of A) $CH_4$ in g $CH_4$ m$^{-2}$, B) $CO_2$ in kg $CO_2$ m$^{-2}$, and C) $N_2O$ in g $N_2O$ m$^{-2}$ for all organic soil collars for the measurement period from February 2022 to January 2023. Shaded areas for $CH_4$, $CO_2$ and $N_2O$ graphs represent the maximum and minimum monthly average fluxes. Blue dashed line below $CH_4$, $CO_2$ and $N_2O$ represent the measured monthly average transect groundwater table depth (WTD) in cm below terrain. Red dashed line shows the monthly average air temperature (TA).**

The average soil GHG fluxes for all collars were summed to monthly transect sums to illustrate long term drivers on the flux magnitude. Overall, monthly sums of $CO_2$ and $N_2O$ emissions increase with temperature and fluxes are highest under deeper WTD, but $CH_4$ net fluxes were less responsive to long term changes in both temperature and hydrology (Fig. 11A-C). Net uptake of $CH_4$ increased slightly with increasing temperature and lower WTD during the spring and summer. With increasing water table and high temperatures in August the soils across the transect turned into a small net $CH_4$ source continuing in fall and winter (Fig. 11A).

For $CO_2$ the seasonal variation was pronounced and closely followed soil temperature until peak values in July for both transect average, minimum and maximum fluxes, respectively (Fig. 11B). From July to August, it was observed that WTD across the transect began to increase again and $CO_2$ fluxes departed from the close relation to soil temperature, indicating an inhibitory role of the WTD in this period, but reaching minimum fluxes in December, corresponding to the wettest and coldest month (Fig. 11B).

Similarly, $N_2O$ fluxes increased with soil temperature reaching peak monthly values in August, corresponding to the period of the year with highest soil temperature and increasing WTD (Fig. 11C). This supports the

promoting role of soil water saturation on the production of $N_2O$ when temperature is favourable for
denitrification. $N_2O$ fluxes reached minimum values in December when WTD and ST were lowest (Fig. 11C).
**3.8.3 Ditch $CO_2$, $CH_4$ and $N_2O$ fluxes**
**3.8.3.1 Time series of raw data of ditch $CO_2$, $CH_4$ and $N_2O$ fluxes**

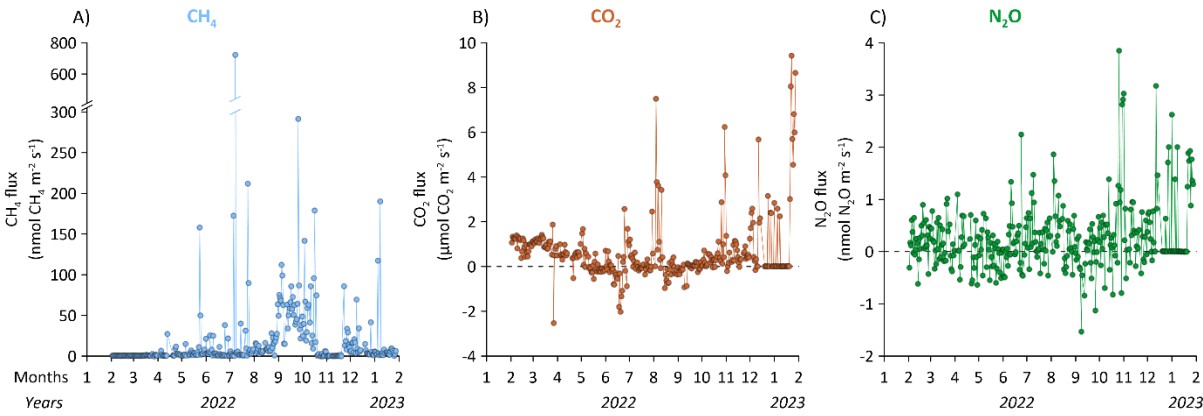


**Figure 12: Daily average time series of net ditch total A) $CH_4$ (diffusion and ebullition), B) $CO_2$, and C) $N_2O$ fluxes at**
**the Vejrumbro site for the measurement period from February 2022 to January 2023.**
Common for all three gases is that ditch emissions are dynamic and net fluxes change from zero to large net
positive or negative fluxes within hours or days (Fig. 12A-C). Compared to net soil $CH_4$ fluxes the ditch can be
considered an emission hotspot at the Vejrumbro site (sum of diffusion and ebullition: 8.3 g $CH_4$ $m^{-2}$ $y^{-1}$), but
fluxes are lower than earlier reports for ditches in other drained wetlands (between 0.1 – 44.3 g $CH_4$ $m^{-2}$ $y^{-1}$)
(Peacock et al., 2021). Methane varies most throughout the measurement period with maximum diffusive flux
close to 700 nmol $CH_4$ $m^{-2}$ $s^{-1}$ and there was a tendency toward consistently higher net $CH_4$ emission from
August to September, becoming close to zero in colder seasons (Fig. 12A). Ebullition of $CH_4$ did occur
occasionally in the ditch, e.g. about 19.3% of flux measurements for the ditch was comprised of ebullitions but
constituted on average only 2.9% of the total $CH_4$ emission (0.24 g $CH_4$ $m^{-2}$ $y^{-1}$) from the ditch which is lower,
but in the same range as a recent estimate from a ditch in a similar drained German peatland (Köhn et al., 2021).
According to the flux calculation methodology, flux separation and extrapolation to daily sums, diffusive fluxes
dominated (6.56 g $CH_4$ $m^{-2}$ $y^{-1}$). However, it cannot be ruled out that the classification as diffusive flux may in
fact be ebullition by nature. It has been suggested that microbubbles resulting from mass transport can resemble
diffusive fluxes in a chamber making it difficult, if not impossible, to fully separate the two emission
mechanisms in a continuous time series if headspace $CH_4$ concentrations do not abruptly increase (Prairie and
del Giorgio, 2013), such as in the example shown in Fig. S4.
For $CO_2$, there was a general tendency towards lower fluxes during the summer months and fluxes increased in
magnitude and variability towards the end of the study period (Fig. 12B). For $N_2O$, the fluxes fluctuated around
zero for most of the study period, except towards the end (December and January) where net fluxes became
positive (Fig. 12C). Compared to the net soil $N_2O$ and $CO_2$ fluxes the ditch fluxes of these gases are low
showing that the ditch is not contributing significantly to the $CO_2$ and $N_2O$ budget at the Vejrumbro site.
Per square meter, the ditch emitted less $N_2O$ (0.41 g $N_2O$ m$^{-2}$ or 2.6 kg $N_2O$-N ha$^{-1}$ y$^{-1}$) and $CO_2$ (961 g $CO_2$ m$^{-2}$
y$^{-1}$ or 2.6 tCO$_2$-C ha$^{-1}$ y$^{-1}$) than the organic soil, but was a hotspot of $CH_4$ emission (8.4 g $CH_4$ m$^{-2}$ y$^{-1}$ or 63 kg
$CH_4$-C ha$^{-1}$ y$^{-1}$) during the measurement period. Although these emissions estimates are lower than previously
reported for ditches in organic soil (up to 44 g $CH_4$ m$^{-2}$ y$^{-1}$) (Peacock et al., 2021). For the ditch $CH_4$ budget,
ebullition only constitutes 2.9% of net $CH_4$ emissions during the study period. This proportion may be
underestimated as the count of ebullition events may have been underestimated (Prairie and del Giorgio, 2013).
**3.8.3.2 Diurnal variability in ditch fluxes**

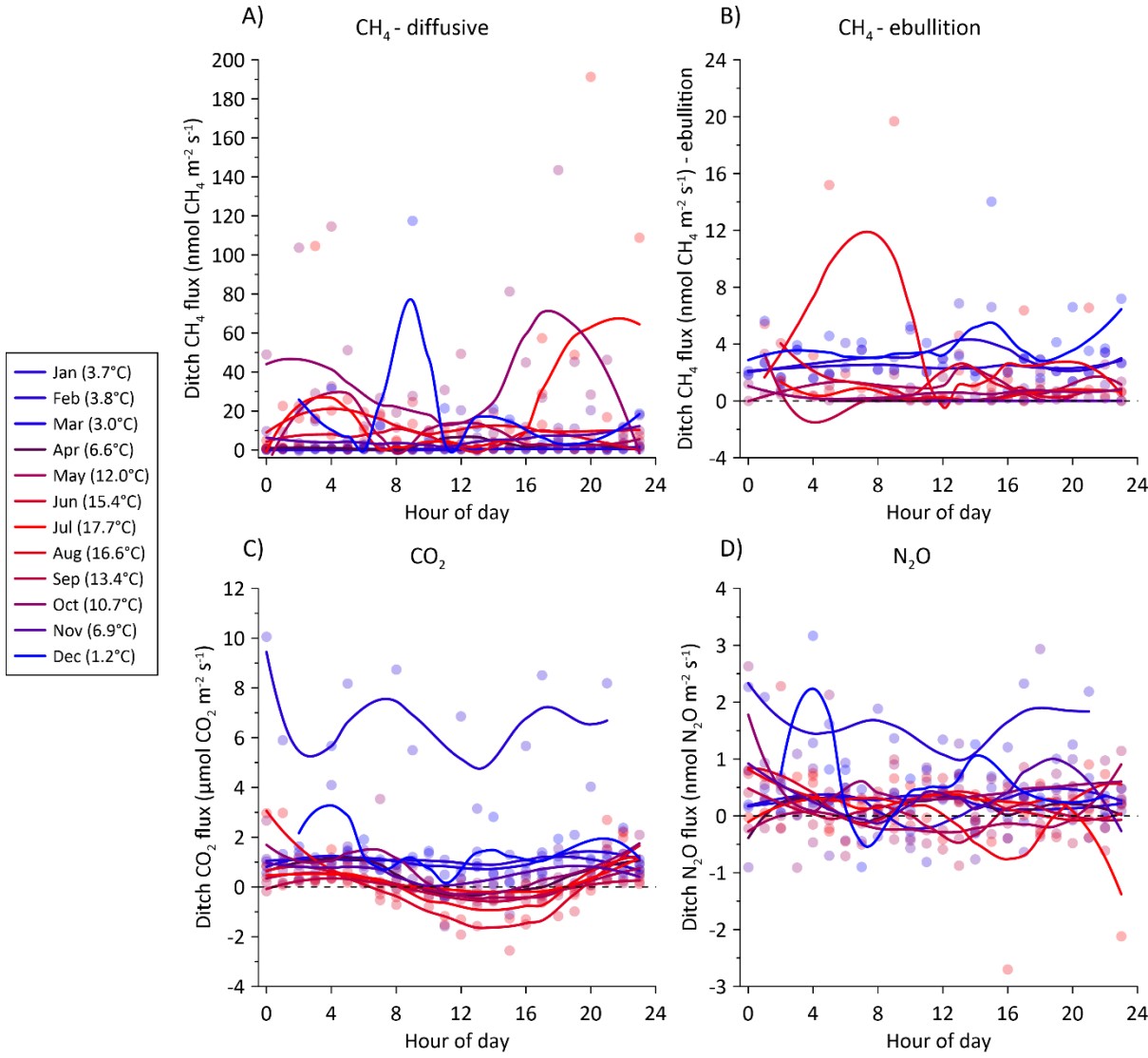


**Figure 13: Average hourly fluxes for the ditch collar of A) diffusive $CH_4$ fluxes, B) CH4 ebullition fluxes, C) $CO_2$, and**
**C) N₂O during a 24 hour period. The fluxes were assigned the hour of measurement during the day and averaged per**
**month. The diurnal variation is split between each month during the 2022-2023 measurement period. Color shade**
**between blue and red corresponds to average air temperature for the specific month shown in parenthesis in the**
**figure legend. Solid lines are loess fits for visualization of the diurnal variation in each month. Note different axes.**
For $CH_4$ fluxes, both diffusion and ebullition, there was no clear diurnal variability in any month (Fig. 13A and
B). This is expected for ebullition emissions which is known to be erratic without any clear diurnality (Sø et al.,
2023; Wik et al., 2016). For net $CO_2$ fluxes from the ditch there was no diurnal variability in colder seasons
(Jan, Feb, Mar, Nov and Dec), but consistent positive net $CO_2$ efflux (Fig. 13C). Diurnal patterns became clearer
with higher temperatures from May to October (Fig. 13C) and in this period $CO_2$ fluxes decreased during the
day to sometimes reach net negative fluxes (net uptake of $CO_2$) during and after midday (Fig. 13C), although the
net emissions were also observed in the daytime period (Fig. 13C). The net negative fluxes can likely be
explained by photosynthetic activity of aquatic plants on the surface of the ditch or by algae in the water column
which was measured due to the transparency of the chamber. Using an opaque chamber instead would likely
have resulted in different net $CO_2$ efflux in daytime. For $N_2O$, the same pattern as for $CH_4$ was observed, where
flux magnitude across the day fluctuated around zero, except for January where $N_2O$ fluxes were consistently
above zero (Fig. 13D).
**3.8.3.3 Monthly variability in ditch fluxes**

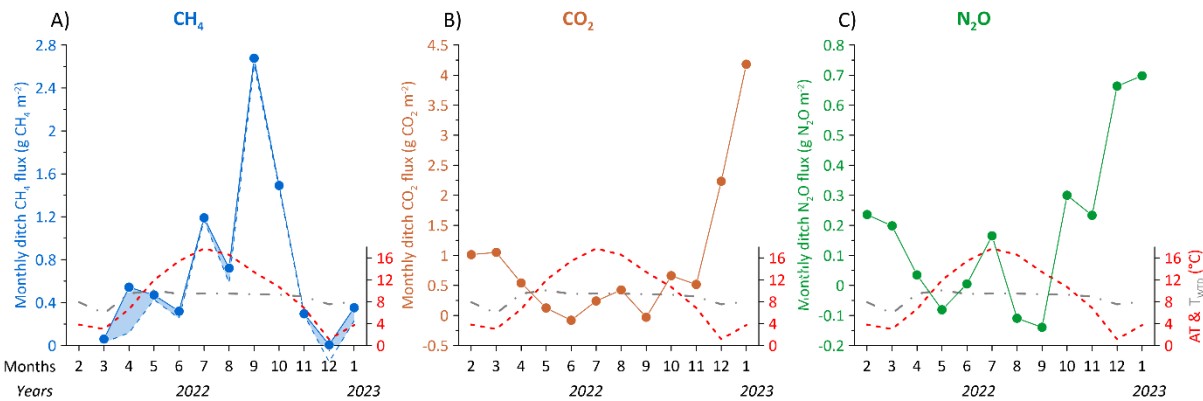

**Figure 14: Monthly summed ditch fluxes of A) $CH_4$ in g $CH_4$ m$^{-2}$, B) $CO_2$ in g $CO_2$ m$^{-2}$ and C) $N_2O$ in g $N_2O$ m$^{-2}$ for**
**the measurement period from February 2022 to January 2023. In A) the blue dashed line is the contribution of**
**diffusive fluxes and the shaded blue area between the full and dashed blue lines represent the monthly contribution of**
**ebullition to the total flux. Red and grey dashed lines show the monthly average air (AT) and groundwater**
**temperature ($T_{WTD}$) in °C, respectively.**
The monthly sums of $CH_4$ tend to increase with air temperature, although peak $CH_4$ emissions (September)
occurred after air temperature peak (July) (Fig. 14A). Diffusive fluxes comprised the major emission pathway of
$CH_4$ in the ditch (between 21% - 99%), with the contribution from ebullition being highest in March (55%) and
April (78%) (Fig. 14A). Water temperature in the ditch was relatively stable throughout the year, varying
between 5.8 – 10.1°C being highest from April to November and lowest from December to March. However,
there is little indication of a direct relation between ditch water temperature and net GHG fluxes (Fig. 14A-C).
For $CO_2$ and $N_2O$, the seasonal pattern is reversed with lowest fluxes during the warmest periods, approaching
net zero or even net negative fluxes (Fig. 14B and C).
**4 Data availability**
Data for this publication is available for download via https://doi.org/10.60612/DATADK/BZQ8JE.
**5 Conclusion**
The dataset presented here is unique for temperate fens and demonstrates the advantage of using automated
GHG measurement systems to resolve temporal and spatial patterns of GHG dynamics in high detail. It

represents a full year of data from 2022–2023 and must be considered specific to this period and the location at Vejrumbro. Consequently, it is expected that the annual budget of all GHGs in other years will likely differ due to varying climatic and hydrological conditions.

Specifically, the dataset demonstrates how temporal variation in soil hydrology and temperature is linked to the temporal variation of fluxes. Interestingly, the temporal variability of GHG fluxes across the transect appears to be lower than the spatial variation highlighting that spatial variability in hydrology and temperature may not necessarily be the best predictor of flux magnitudes across the transect. The cause of spatial variability in GHG fluxes remains unresolved and does not clearly link directly to either water table depth (WTD), soil temperature, or soil/groundwater chemical parameters.

The initial harvest and herbicide application represent ecosystem disturbances that could potentially alter soil biogeochemistry. However, these were conducted months prior to the start of flux measurements, minimizing the direct effect of herbicide. Continued plant removal from inside the collars was necessary for flux measurements, meaning the fluxes may only be regarded as net soil GHG fluxes and not representative of net ecosystem exchange. Excluding vegetation likely influenced measured fluxes of soil respiration (e.g., excluding root exudates) and reduced plant-mediated $CH_4$ and $N_2O$ emissions, potentially also reducing interannual variability.

*Carbon dioxide fluxes*: The magnitude of annual cumulative $CO_2$ fluxes is in the same range as other studies of temperate fens. Temporal variability is largely governed by the seasonality of WTD and soil temperature ($T_{soil}$). Soil $CO_2$ fluxes showed diurnal variability with higher fluxes during midday, where the amplitude between night and day was augmented with $T_{soil}$.

*Nitrous oxide fluxes*: Cumulative soil $N_2O$ fluxes exceed previously reported values for temperate fens at the Vejrumbro site and others. Unlike $CO_2$, $N_2O$ is emitted largely in pulses related to rapid fluctuations of WTD, which increase in size with $T_{soil}$, indicating a seasonal regulation of $N_2O$ production by temperature. These measurements suggest an important but difficult-to-capture dynamic of $N_2O$ in peatlands, where hot moments during warm periods determine most of the annual emissions. Soil $N_2O$ fluxes also showed diurnal variability similar to $CO_2$.

*Methane fluxes*: The peat soils across the transect were insignificant sources of $CH_4$ during the measurement period. This could be linked to deeper WTD (20–40 cm) during summer, a cold wet winter, and the presence of alternative electron acceptors ($NO_3^-$, $SO_4^{2-}$, and $Fe^{3+}$), which provide suboptimal conditions for $CH_4$ production. Vegetation removal may have further impeded $CH_4$ emissions by restricting plant-mediated pathways. Soil $CH_4$ fluxes did not show diurnal variability.

The ditch at the transect was a net source of both $N_2O$ and $CO_2$, but at magnitudes 27 and 4 times lower than the soil GHG fluxes, respectively. It acted as a $CH_4$ source, comparable to other ditches in temperate fens. $CH_4$ was emitted mostly through diffusive emissions from the water surface, with occasional observations of ebullition.

This dataset provides a unique opportunity to test hypotheses regarding spatial and temporal patterns of GHG emissions and their drivers in peatlands. It supports the development of models that predict soil GHG fluxes in response to soil temperature and hydrology (WTD), aiding in the prediction of reliable budgets for locations

beyond Vejrumbro. We intend to publish this dataset to the research community so that experimentalists and
modelers can use it to explore basic hydrological and thermal regulation of GHG fluxes and develop predictive
models for spatiotemporal variability.

**Competing interests**

The authors declare that they have no conflict of interest.

**Author contributions**

JRC, PEL and KSL designed the experiment and carried them out. ASN performed flux calculation and quality
checking. RJP and PEL installed the equipment for groundwater measurements. All authors contributed to
writing of this manuscript.

**Acknowledgements**

The measurements are the results of the RePeat (grant nr. 33010-NIFA-19-724), INSURE and ReWet (grant nr.
5229-0002b) projects hosted by University of Copenhagen and Aarhus University. ReWet is part of the Danish
roadmap for research infrastructure funded by The Danish Agency for Science and Higher Education. INSURE
was part of EJP Soil and received funding from the European Union's Horizon 2020 research and innovation
programme under the grant agreement no. 862695.

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
