# Peer review of "A full year of continuous net soil and ditch CO2, CH4, N2O"

_Earth System Science Data, 2025_

## Referee Comment (RC3)

Review of:

**A full year of continuous net soil and ditch CO2, CH4, N2O fluxes, soil hydrology and meteorology for a drained fen in Denmark**
Annelie Skov Nielsen, Klaus Steenberg Larsen, Poul Erik Lærke, Andres Felipe Rodriguez, Johannes W. M. Pullens, Rasmus Jes Petersen, and Jesper Riis Christiansen

This data paper is a useful contribution to the study of soil greenhouse gas fluxes in peatlands, because it presents data from a novel system of automatic chamber greenhouse gas measurements covering a transect with a large number of collars, a relatively high data frequency and a full year coverage. However, I have two serious objections to this paper that should be addressed properly before final publication. Therefore I recommend publication, but with major revisions. My main objections are:

1. A lack of ancillary soil and vegetation data. There is practically no data on soil and vegetation included, except for a short mentioning of peat soil. Neither it is clear which of the cited references gives adequate information on soil conditions of the transect. If these data are to be be used by other researchers on ecosystem greenhouse gas fluxes, one would at least expect a basic description of a soil profiles at the site, or a borehole transect, and some basic soil and water chemistry data.

2. The measurement procedure, that entails removal of vegetation. The procedure of removal of vegetation has been common in the past but is increasingly abolished because of the intense intertwining of vegetation, microbial community and soil processes that generate greenhouse gas fluxes. Removal of vegetation (including the application of herbicide in this case) is a large disturbance of this system, with questionable results. It introduces artefacts that are poorly quantified for $CO_2$ fluxes since labile carbon pools in the soil are affected.
In the case of $CH_4$ fluxes, the main supply of labile carbon for methanogens is reduced, and the main transport pathway of $CH_4$ from soil to atmosphere (by plant aerenchyma) is destroyed. Therefore it likely leads to much lower fluxes of $CH_4$ compared to those in an undisturbed system, resulting in data that cannot be compared to those of other sites – if not simply flawed. For details, see the comments below.
The authors should state clearly in the abstract that vegetation removal has been applied. Furthermore, they should discuss properly what effects this may have had on the fluxes that they have measured.
Even if this is a data paper only, reflection on the complexity of the system that you have measured, and on the effects of your measurements on that system, is necessary.

Minor comments concern mostly the quality of figures and their captions, and questions on the operation of the automatic chamber system.

**Detailed comments on the paper.**

Section 2.1 Site description: This is disappointingly incomplete. Not any information is given on the soil profile and it lateral variation along the transect, while this could have been checked with a few hand augerings. What is the peat stratigraphy, are there any sand or clay layers in between or on top of the peat? What is the peat type, its decomposition grade, loss on ignition? Any information about soil water chemistry, for instance the presence of anaerobic electron acceptors that influence the redox potential and methanogenesis? What is the variation of the vegetation along the transect? *Juncus effussus* and most grasses differ strongly in the characteristics of their root system and methane transport characteristics. All this is information that any user of your data would want to know.

Line 94 – 101, caption Fig. 3: What are the instruments in the lower right corner of the figure?

Figure 4: The figure is not very informative (except on the surface topography and placement of the collars) and the caption is confusing. The vertical profile has two colours, brown and dark grey, which suggests some sort of stratigraphy. However, the brown colour is marked as 'transect surface', but apparently it indicates the soil above the minimum water table depth. The lower depth of the peat is not indicated. Information on the peat properties and its variability (e.g. the presence of clay/sand layers) is lacking. This would be very useful information for users of the data.

Line 170 – 173:
This procedure of killing vegetation by harvesting, and application of a herbicide, attempts to reduce the effects of vegetation respiration and to measure the 'true' or 'net' soil GHG flux. However, it introduces other artefacts that are poorly quantified, in particular for the $CH_4$ fluxes. For measuring of $CO_2$ fluxes from the soil it often has been done with the purpose of reducing $CO_2$ respiration/uptake by plants. Because of the artefacts it introduces, alternative approaches have been developed that leave vegetation intact and separate soil and vegetation components of the flux by modelling (e.g. Boonman et al., 2024). For the $CH_4$ fluxes it may have resulted in serious underestimation of the fluxes.
The statement that the fluxes after removal of the vegetation represent the 'net' soil greenhouse gas flux is invalid without specification what is actually meant by 'net flux', in particular when it is not explained which soil carbon pools are assumed to contribute to to this net flux. It may at best approach the soil $CO_2$ flux (with an unknown error or bias) and likely severely underestimates the $CH_4$ flux. In general, $CO_2$ from the decomposition of recently produced labile carbon, and that from older soil carbon (e.g. the peat carbon pool) is difficult to separate in surface flux measurements. Vegetation removal does a poor job in that, because most the root mass often remains behind, will be active, and also affects the microbial population.
Besides these caveats, I also wonder if vegetation removal is a specific requirement of this automatic chamber system. Does vegetation hamper a leak-free placement of the chamber on the collars with this system?

A motivation why this procedure has been applied, and a discussion of the caveats listed below is necessary. I suggest to do this in a separate Discussion section. Also, the vegetation removal procedure itself should be mentioned clearly in the abstract, for potential users to judge wether the data are suitable for use.
Drawbacks of vegetation removal:
1. The soil greenhouse gas flux in an undisturbed ecosystem is the sum of all peat and other organic matter decomposition. The 'other' being various forms of recently produced, usually labile carbon, produced by the vegetation root system and litter decomposition. Removal or decrease of one carbon pool may strongly affect the measured fluxes, in particular when it is not known quantitatively what has been removed. Furthermore, labile carbon interacts also with stabile carbon decomposition via the priming effect. This may enhance stable carbon decomposition (e.g. peat decomposition) in the presence of labile carbon. Therefore, it should be specified which carbon pools are considered to be included in the flux measurements (peat, older humic matter, recently produced organic matter, labile or stable?), and what effects the measurement procedure has on $CO_2$ emission from these pools. If actual data collection, e.g. root mass, is not available, the authors could at least consult literature from other sites on that.
2. Glyfosate is known to affect soil faunal and soil microbial respiration (e.g. Nguyen et al., 2016). The application of this herbicide will have influenced the measured fluxes to an unknown extent.
3. As the need for very frequent removal of living vegetation during the experiment testifies, the root system in the soil remained active, producing labile carbon and adding a vegetation and labile carbon respiration component to the fluxes. Therefore, vegetation removal still does not remove vegetation effects.

4. Since detection of $CH_4$ emissions is included, you are removing one of the main transport mechanisms of $CH_4$ from soil to atmosphere: the transport via plant aerenchyma (e.g. Vroom et al., 2022). Moreover, the main source of carbon for methanogens is labile carbon compounds produced by plant roots. The low $CH_4$ emissions therefore may be flawed and not represent normal ecosystem or soil $CH_4$ fluxes. On peat soils with approximately similar water table variation and vegetation, significant positive $CH_4$ fluxes were measured with manual and automated chambers (Hendriks et al., 2007; Lippmann et al., 2023).

Furthermore, you say that you removed vegetation with a minimum of 7 days. What was the (probably higher) vegetation removal frequency in the spring and summer period? This is highly important given the rapid vegetation regrowth in that part of the year.

Line 179: How does wind speed affect the operation of the system? How reliable is it at higher wind speeds?

Line 180: How certain can you be that rapid vegetation growth near the collar does not affect the airtight connection of chamber and collar? For instance, leaks may result from high grass getting between the chamber and the gasket during windy conditions.

Line 182: If a fan was not installed in the chamber, what is the air flow provided by the main pump, and is it sufficient to flush the chamber?

Figure 5: Can the upper photos be made sharper or larger, providing more detail on the chamber construction? Eventually, provide them in a supplement.

Line 232: Check the sentence starting with "If the relative SE…", the part "than 100%" appears to be misplaced.

Line 238: During the measurement time the temperature in the transparent chamber can rise in a matter of a minute in sunny weather. Is there no temperature sensor inside the chamber to detect this effect and correct for it?

Caption Fig. 7. Good point, but how sensitive is the system to bubble fluxes induced by the chamber lowering? Was the collar anchored somehow in the subsoil of the ditch to prevent disturbance?

Line 383: variability in soil water content. Again, it is disappointing that so few soil data is included. Could there be differences in water seepage from higher ground (which is to be expected given the topography) or does soil cracking in dry periods occur, influencing the SWC?

Line 398 – 402. Again, the large spatial variability should be no surprise. Unfortunately, any information on soil variability is missing. For instance, I would have expected information on the soil carbon content, which is an important predicting variable in $CO_2$ fluxes from soils. Although the above-ground vegetation is removed, there is still root mass present that produces labile carbon; root density also adds to the spatial variability. This kind of data would be very useful for other users of the data.

Line 424. Interesting to see these bursts. As mentioned above, this could also be an artefact of your methodology. It excludes plant emissions, which is usually a major $CH_4$ emission pathway (Vroom et al., 2002). By artificially removing the plant flux, a buildup of $CH_4$ concentration in the soil could induce a burst-like emission pattern.

Figure 16 and Figure 19. This figure is difficult to understand, information in the caption is ambiguous.

You have only 5 measurement points per day, but there are more observation points. The caption suggests that the points are based on a grouping of all soil collars together. Is this over one day, if so, which days? Or over an entire month, as the reference to the figure legend suggests? In that case I would have expected way more data points in the figure, unless it is a monthly average per collar. In short, be clear how your data have been grouped. This is also important for understanding sources of variation in the data. Grouping of all collars in one day introduces also spatial variation, next to temporal variation.
Next, the colour scale of the legend is not very distinctive by choosing only shades of blue and red. Better include other colours as well, which makes the data from different months more distinctive.

Line 443-444. The lack of diurnal variation for $CH_4$ may also have been caused by removing the vegetation. Plant fluxes of $CH_4$ tend to have diurnal variation, driven by solar radiation (Vroom et al., 2022).

Line 540. 'it cannot be ruled out that living roots inhabited the soil below the chambers'. You can be quite sure about that if you have to clip the vegetation frequently!

Line 547-550. There could be other electron acceptors inhibiting methanogenesis, for instance $Fe^{3+}$, sulfate. Here again, some information on basic soil and water chemistry could have been helpful for users of the data.

Table 3. Mention the source of the GWP factors used here.

Section 5 Conclusions: The causes for spatial variability of the GHG fluxes is unresolved – but that is not surprising given that any information on soil variability is lacking. Fig. 15 suggests that the spatial variability is larger than the temporal variability on these closely spaced collars, which would be an interesting conclusion.

Line 578-580: The low $CH_4$ emission is attributed to low water table and a cold wet winter. However, the huge elephant in the room here is the potential effect of vegetation removal on the $CH_4$ fluxes detailed above. At other peat sites with similar water table and vegetation that I have measured myself, persistent positive summer $CH_4$ fluxes occurred (Hendriks et al., 2007; Lippmann et al., 2023). Neither, alternative explanations for the low emissions are considered, such as the presence of other anaerobic electron acceptors (e.g. sulfate reduction) that maintain a too high redox potential for methanogenesis? This is mentioned elsewhere in the article for $NO_3^-$, but not considered here.

**Supplement.**

Missing: collar numbers at each graph. What do the ticks on the horizontal axis represent? First day of each month, midpoint? Day numbers would be more informative on the horizontal axis!

**Data.**
The data representation is largely correct. However, for greenhouse gas fluxes it would be useful include the standard error of the flux calculation method that is applied. This would allow data users to apply additional quality checks.

**References.**

Boonman, J., et al. (2024):Transparent automated CO2 flux chambers reveal spatial and temporal patterns of net carbon fluxes from managed peatlands. *Ecological Indicators, 164,* 112121.

Hendriks et al., (2007): The full greenhouse gas balance of an abandoned peat meadow. Biogeosciences, 4(3), 411-424

Lippmann et al. (2023): Peatland-VU-NUCOM (PVN 1.0): using dynamic plant functional types to model peatland vegetation, $CH_4$, and $CO_2$ emissions. *Geoscientific Model Development, 16*(22), 6773-6804.

Nguyen et al. (2016): Impact of glyphosate on soil microbial biomass and respiration: a meta-analysis. Soil Biology and Biochemistry, 92, 50-57.).

Vroom et al., (2022): Physiological processes affecting methane transport by wetland vegetation, Aquatic Botany 182:103547

Ko van Huissteden
June 2025

---

## Author Comment (AC1)

**Detailed replies to reviewer comments/questions**

**Reviewer #1 Daniel Epron**

With their manuscript, the authors would like to share a full year of data on greenhouse gas fluxes from soil and ditch in a drained fen in Denmark. This is really kind of them, especially since the dataset appears to be of very good quality and obtaining it must have involved both financial and human effort. It is highly respectable, but surprising (for me at least) that they don't seek to promote it with a conventional article before sharing it with the rest of the community. I am not used to reviewing data papers and hope my comments will be helpful nonetheless. Author: Thank you for the kind words.

I have only one major concern related to the discarded data. Discarding non-significant regressions as explained line 238 is a problem. If the flux is almost 0, the slope is also almost 0 and the regression, by nature, is non-significant. If a significant number of low fluxes were discarded, then means and cumulated fluxes used for annual budget in Table 3 for example are over estimated (in absolute value). Similarly, "wrong windows" (lines 251-259) can be repositioned. Of course, this takes time, but it is worth doing.

Author: We fully agree with the concern of the reviewer here and from the current text our methods of discarding noisy and low fluxes can be misunderstood. We did not use R2 as a parameter to determine fluxes, but instead used RMSE, g-factor and the relative error of the slope (SE/slope) as outlined in lines 227-235. We have reformulated the text in lines 238 to clarify that these low fluxes were discarded as it was not possible to visibly detect whether there was a flux or it was because the chamber had malfunctioned. Therefore, discarding these doubtful fluxes is a conservative approach, as including erroneous small fluxes would also bias the means and cumulated fluxes. The new text in lines 311-315 emphasizes this: "At low flux levels non-significant fluxes were discarded as it was not possible to visibly detect whether there was a flux due to high noise-signal ratio of the analyser and/or it was because the chamber had malfunctioned. It is acknowledged that discarding low fluxes can bias annual means and cumulative values, but the data quality did not allow us to determine whether the flux measurement was performed correctly and hence a conservative approach was chosen as including false low fluxes would also bias the data set."

Regarding the "wrong windows". Unfortunately, it is prohibitive in terms of time (resources) to refit "wrong windows" as this will have to be done manually, specifically by identifying these individual flux measurements in the millions of lines of data (in total we have around 31000000 lines of data for each gas). It is our assessment that we will only gain a few percent added data coverage which does not stand against the excessive resources we will spend time wise. We have improved the detection of fluxes in our new deployments of the SkyLine based on these experiences described here.

Minor comments

L25: replace by "flux was more dynamics" Author: Done

L43: "changes" (plural) Author: Done

L59: explain why night-time fluxes are expected to be overestimated Author: we added the following text: "due to atmospheric stratification that disturbs the steady-state diffusion gradient between soil and the atmosphere". We prefer to keep it to this simple explanation as

going in to depths with a physical explanation is unnecessary in our view. The paper by Brændholt et al. 2017 explains this excellent.

L166: provide the length of the long tube (important that the colleagues that will work with these data understand well how it was measured) Author: the lengths of the tube in the ditch and the collar on top has been added to the text

L166: "On the top" is duplicated. Remove one. Author: Done

L173: clarify "thus" and the link between "net flux" and weed killing Authors: We agree that the current explanation to this important disturbance was inadequately described. We have expanded on this section of the paper as it has also been pointed out by reviewers 2 and 3 as important to explain. We have included the timing of the glyphosate application prior to flux measurements, a notion of the half life of glyphosate and explained the plant removal in more detail. We have changed the paragraph now in lines 134-139: "Plant removal from collars is considered a common practice to isolate net soil GHG fluxes as the aboveground autotrophic respiration is removed. Since the individual collars were not trenched it is unavoidable to include belowground autotrophic respiration from plants growing adjacent to the collars. To avoid excessive disturbance of the site we did not remove these roots."

L241: Use "annual cumulated fluxes" instead of "annual budget" Author: Done

Fig 9,10: thicker green line will be appreciated Author: Done

L345: was -> were Author: Done

Fig 11, 18B, 18C: a horizontal line at intercept 0 will help Author: Done

Section 3.6 seems oddly structured, starting with the annual cumulated fluxes, followed by individual measures (5 per day) and then the monthly sum Author: we understand the concern of the reviewer. However, our idea was here to organize the section into; spatial, temporal (daily, monthly) and then finishing with annual cumulated fluxes. We understand that this will lead to a certain overlap in data representation, especially between 3.6.1 and 3.6.4. We would prefer to keep this organization as we do not think it impedes the interpretation of data. Also, both R2 and R3 did not point to this organization.

**Reviewer #2 Judith Vogt**

I would like to thank the authors for compiling this extensive dataset and corresponding manuscript. I think the dataset could be very valuable to the research community. Before publication, I think both the dataset and the manuscript need revision. Author: Thank you for the positive reply.

General comments:

The structure and content of the manuscript reflect that of a classic research article, but the authors chose ESSD for publication. Given the focus on datasets in ESSD, I strongly recommend to revise the datasets to make them easy-to-use and comprehensive for potential users.

I find a bit concerning that the site experienced different levels of disturbance and wonder if that may affect the potential for modelers to use this data? Some discussion could be added in the manuscript, and possibly convincing arguments why this dataset would still be valuable for further use in models and experiments. Author: This is a valid concern and is shared by R1 and R3. We realize that our initial description was too vague and we have added missing details on the glyphosate addition and vegetation removal in section 2.6.1.

Generally, the impact of this study could be emphasized further and throughout the manuscript. Some more in-depth analysis would be nice regarding annual budgets of the site, and driving factors of GHG fluxes, for example, with correlation analyses or similar. Author: It was never our intention with this paper to test hypothesis or infer potential causal relationships to environmental drivers, but to publish the dataset. We trust that the dataset holds many possibilities for alternative analyses, e.g. time series analyses, correlation/statistical modeling, biogeochemical model test/development. We consider the publication of the data set as service to the community to focus future potential publications/studies on focused use of data for analyses without tedious description of the entire methodological setup. Regarding the impact of the study for wider peatland biogeochemical research, the data is confined to one site which, by definition, limits its broader representativeness. We wish to refrain from performing correlation analyses of driving factors as this would increase the length of the manuscript considerably and require to refocus the entire manuscripts towards hypothesis driven approach.

Dataset:

- I would recommend to use csv files rather than excel if the data is meant to be broadly used. Author: this is a good suggestion and we will change the data format to CSV if the revised version of the manuscript is accepted for publication.

- Were the measurements split up in different files because of varying temporal resolutions (I don't see the need to separate them by figure)? If so, it might make sense to leave them in separate files. In that case, please use uniform column names, e.g. TIMESTAMP instead of Date, DateTime or Time and avoid several columns with the same name. Otherwise, it would also be nice to have all information in a single file, or at least have one file with flux measurements and accompanying WTD, etc. Author: We understand the value of merging flux values with hydrological and climatological data. However, as pointed out here by the reviewer the data files were split due to different temporal resolutions of the different data types. We appreciate the reviewer can see the sensibility in maintaining split files based on measured parameters and we

have accordingly streamlined the Date, DateTime, Time to one common name – TIMESTAMP – as suggested.

- I would recommend to give latitude and longitude of the locations in each file. An indication whether the collar is soil or ditch might also be helpful as well as any other indicators that show differences among the collars (maybe elevation?). Author: We are unsure of the value of adding lon/lat + elevation for each collar in the data files as this will increase the size and complexity of the data files and would therefore like to refrain from adding this information. Instead, we made the file "VB SkyLine2D transect Figure 3.xlsx" to contain general information of the placement of the transect, that can be easily cross referenced with the collar number in data files, but we also now realize that the exact lon/lat position of each collar is missing in this file as pointed out. We have added the lon/lat information in the "VB SkyLine2D transect Figure 3.xlsx" in columns O, P file as well as indicated whether it is peat or ditch. The elevation in meters above sea level is already provided in column P of the current "VB SkyLine2D transect Figure 3.xlsx" file.

- I don't see the ebullition fluxes in the dataset. Please clarify how they are reflected in the CH4 column and ideally split up into diffusion and ebullition. Author: We assume you refer to the file "VB GHG fluxes Figures 9 - 15.xlsx". In column D, F and H of the file is indicated whether the flux was calculated as HM: Hutchinson-Mosier (non-linear), LM: linear regression or as ebullition. Section 2.7 and 2.8 details the calculation of diffusive (HM and LM models) and ebullition fluxes, respectively.

Specific comments:

Abstract:

- Especially in the abstract and introduction, I suggest to clarify the directions of fluxes, e.g. "soil-atmosphere/ditch-atmosphere fluxes" and "emissions/source to the atmosphere". Author: We agree that the language regarding the fluxes and distinction between soil and ditch in the abstract was unclear and we have clarified. In the first paragraph of the introduction we have added more detail on the net emission/uptake in relation to GHG fluxes and the relation to net emission/sink of atmospheric GHGs.

- note that the doi of the dataset should be given in the abstract for ESSD Author: Done

- add numbers of average fluxes for CH4 and N2O as is done for CO2, and add standard deviations or similar Authors: numbers of $CH_4$ and $N_2O$ are already mentioned in the abstract (e.g. 42515 and 44099, respectively). We have added average $CO_2$, $CH_4$ and $N_2O$ for all soil and ditch fluxes.

l. 26: replace "lead" with "led" Author: Done

l. 27: clarify what is meant by annual budget – N2O or GHG budget? Author: clarified as "annual net budget of soil $N_2O$"

l. 30: remove "with expectations" Author: Done

Introduction:

- since it is suggested to use this dataset for models, it might be worth to search the literature and include a paragraph about how/why this dataset is currently lacking and would be a valuable addition to the modelling realm Author: we have expanded on the need for this dataset in the introduction

- the second paragraph only mentions a single measurement unit. It might be worth to mention a few others and make the advantages of the system used in this study a bit clearer. Author: we added mentioning of the Eosense eosAC-LT/LO system, deploying different chamber designs than LiCOR.

- two paragraphs of the introduction focus on processing methods, although I don't think that is part of the main points of this manuscript. Therefore, I would recommend to shift the focus a bit and make the impact of this manuscript a bit clearer. Author: We agree that we do not discuss the implications of different data processing on fluxes in the manuscript and we only use one approach with the goFlux script. We reduced these two paragraphs to one and expanded on the impact of our data set in a broader context in the last paragraph. We like to refrain from excessive speculation regarding fluxes for this particular site to environmental drivers as this is an explicitly stated aim of why we publish to paper that we seek collaboration with other researchers to achieve this deeper insight. We truly believe that sharing the data in this way will advance the interpretation.

- in the Methods you describe that the site went through a chain of disturbances/changes which likely affect GHG fluxes. Therefore, this should definitely be made clearer in the introduction as well. Author: We are not entirely sure if the reviewer here means 1) the initial disturbances of cutting, application of glyphosate and continued aboveground plant removal or 2) the rewetting that have taken place at the site. However, we believe the reviewer refers to the initial plant disturbances and have expanded on the description of this by adding text on how the initial harvesting, subsequent application of glyphosate and continuous removal of vegetation throughout the measurement periods in lines 120-139 where we also discuss how this may have impacted fluxes and what our approach presents of limitations towards establishing a true net ecosystem GHG budget. We are fully aware of the limitations with our setup and believe they are communicated, but currently there are no ideal chamber systems that can measure the full GHG balance without presenting some level of disturbance.

l. 48: add "a", i.e. allow for a standard number of… Author: Done

l. 65: in Figure 3, the distance is 30 m, not 24 m – please revise. Author: In the caption for Figure 3 it is stated that the transect with the collars is 24 meters long and visually it is shown that the distance between the towers is 30 meters. This is also described in lines 190-191.

Materials and Methods:

l. 73: add "in Denmark" and the latitude/longitude of the site Author: done

l. 79: replace "makes out" with "forms" Author: Done

l. 86/Figure 1: The scale bar on the lower left is not visible. Author: Figure 1 has been updated

l. 90/Figure 2: This figure could be shifted to the appendix or be removed. Author: Figure 2 has now been moved to Figure S1

l. 93/Figure 3: Lines and text not visible very well, please increase size and visibility, e.g. by adding white background to black text. "30 m" could be red for clarficiation. Author: the figure has been updated to improve readability

l. 95-97: unclear what the JB numbers are without reference Author: this reference has been removed as it was irrelevant and a section on peat soil characteristics has been added as per request of R3 (lines 143 – 161). The JB numbers refer to a specific Danish soil classification for agricultural soils.

l. 98: yet another transect length is given. Please stick to one throughout. Author: sorry for the confusion. It was a left over from earlier. The length of 24 meter is now used consistently

l. 99: specify analyser, e.g. "greenhouse gas analyser"? Author: we have streamlined the wording regarding analyser and mentioned GHG before analyser throughout the manuscript.

l. 100-101: does the elevation refer to above sea level? Please specify. Remove repetition of "along the transect/across the transect". Author: clarified

l. 102: replace "an N-S" with "a north-south" Author: done

l. 103: briefly mention what the SkyLine2D system is, add a reference or manufacturer Author: done

l. 104: clarify whether the the pallet tanks were simply used to stabilize the system Author: this has now been clarified

l. 109: replace "was" with "were" and remove repetition of "along the transect" Author: done

l. 110: replace "farmer's field" with "agricultural field" Author: done

l. 118-119/Figure 4: The peat depth is not indicated in the figure, so I think the last sentence in the caption could be removed. Author: done

l. 120: I don't find the section heading "Data variables" fitting here since the variables are not clearly presented, but it is rather described how they were measured as would be done in the Methods section of a classic research article. Author: we have given this section a new title "Overview of time series of GHG fluxes, soil temperature/moisture, air temperature, wind direction and groundwater level". We would prefer to keep this table

l. 121: inconsistency with length of dataset – previously it was 12 months, now 13. Author: yes, this is a mistake. This has now been clariid both in the text and table 1.

l. 123: Please indicate whether there was any snow cover during the study period at some point in the manuscript. Author: there was a shorter period between December, 7th and 19th, but the depth was not estimated. This information has been added.

l. 124: revise dates to include a comma between day and year and stick to the same date format throughout the manuscript (also in caption of Figure 12, for example) Author: done

l. 127/Table 1: The last column is missing a header – should it be "Data availability" or similar? The footnote 2 could be mistaken with exponent 2 – maybe consider choosing a different character such as asterisks instead. Author: we have clarified that the columns containing 2021, 2022 and 2023 mark data availability of the listed variables in the caption and by adding a top row to the table with the word "Data availability"

l. 133-135: Worth mentioning here that these measurements were only conducted at specific collars. Author: done

l. 143: replace "collecting" with "measuring" Author: done

l. 154-160: While I don't think section 2.5 is essential for the manuscript, it may be helpful for others in the research community. As a note, links should include the date of last access and abbreviations should be explained (such as SigFox). Authors: We agree that this is marginal informative, but it can be helpful if people want to replicate to know the reference. The names LoRaWAN and SigFox are standard names for these protocols and as such are not abbreviations.

l. 161: I think section 2.6 should come earlier in the manuscript since the SkyLine2D system has already been mentioned several times before without further explanation. Also, the distinction into subsections 2.6.1-2.6.3 does not seem overly helpful and they could be merged into one since they all describe the measurement setup. Note that at the beginning of a sentence, numbers should be spelled out. A little restructuring of section 2.6 might help to chronologically answer the questions: What was measured and why? How were things measured? I think some information is very detailed (e.g. some data processing steps in l. 219-226) and could be moved to the appendix to shorten this section a bit and focus on the most relevant information. Author: We agree the SkyLine2D concept should be introduced earlier which is why we have moved some of the text from 2.6 to the site description – see lines 184 - 210. We merged sections 2.6.1 to 2.6.3 in to one section 2.6 to improve readability, but maintain the argument of including it in the main manuscript as it is the GHG flux methods that is at the heart of creating a high quality data set. Similarly, with sections 2.7 and 2.8 detailing the flux calculations. We think this is needed in a data descriptor paper to fully describe how the data was created. In this case it relies heavily on the measurement system (2.6) and how the resultant data was processed to fluxes (2.7 and 2.8). It is too rare that scientific papers take the time to fully explain the technical details in flux measurement and system design. Here we believe we have an opportunity to do this, so subsequent papers do not need to go through the same tedious details. We therefore argue to maintain the level of detail as a service to the user of these data.

l. 169: replace "hit" with "sat on" – does this mean that if the water level was low, air could have entered through the holes? Author: corrected. The holes in the pipe in the ditch were placed lower than the minimal observed water level in the ditch. This information has been added in section 2.6 first paragraph.

l. 170-173: I think it would be worth to dedicate a subsection to summarize all the disturbances

the site faced for clarification. Was the addition of Glyphosate done to reflect common agricultural practices? Or rather to get the true soil flux? Could this treatment have affected GHGs? Author: We agree with this suggestion and we have added in the site description a comprehensive description and discussion (lines 106-139) of the main disturbances on the site, e.g. glyphosate addition and plant removal. It is here detailed that we could not, because of chamber dimensions, include aboveground living plants as they grow too big here. Therefore, our strategy was to focus on net soil fluxes, which necessitated removing vegetation initially and continuously. In this text we also acknowledge the drawback of these practices. However, they are routinely practiced in chamber flux research due to inherent limitations of the technique to represent the entire ecosystem. We acknowledge this and argue that our study is on the mechanisms of soil GHG dynamics rather than the net ecosystem exchange for which inclusion of plants is needed.

l. 185-189: I think this could be removed or are these variables relevant and given in the dataset? Author: Agree and we removed the text.

l. 190/Figure 5: This is a nice figure – make sure all text and numbers are visible. I think it could be moved to the appendix since it shows details of the measurement setup. Author: agree. This figure is now Figure S2

l. 206: replace "hz" with "Hz" Author: done

l. 207: The precisions seem to be for 5-minute intervals? If so, please indicate. Author: done

l. 214: remove quotation marks Author: done

l. 215: sentence could be removed Author: removed

l. 217: remove "the procedure outlined in" Author: done

l. 218: I think this sentence could be removed, or correct to "converted to micromole per mole" Author: removed and replaced with the unit of the fluxes for $CO_2$, $CH_4$ and $N_2O$

l. 228 and 233: Why "at time zero"? The slope would cover a range of timesteps. Author: for the HM calculation the slope is time dependent, e.g. you want to estimate the pre-deployment flux. The flux at time zero is the best estimate for this flux. This has been the convention since the HM paper was published. For the LM the slope is time-independent and hence represents the pre-deployment flux.

l. 237: How were you able to detect mechanical malfunction? Author: There are built-in features in the SkyLine ensuring the system to stop running when it encounters mechanical anomalies, for example due to that the chamber did not lower or raise properly or that the trolley missed the end stop. This is to avoid damage to the equipment. This has been clarified now

l. 238: What do you mean by "non-significant"? Is this based on a p-value? Please indicate. If low fluxes were generally removed, does that mean that you overestimate fluxes? Author: Agree, this was unclear and in fact incorrect. R1 asked a similar question. We refer you to comment #2 to R1.

l. 241-246: I am surprised that this simple approach to estimate annual budgets was chosen with this high-resolution dataset. Was there a reason why the daily values were not summed over the year instead of taking an average? At least a few different estimates using different simple methods to determine annual budgets could be presented. In my opinion, this point could be a strength of this manuscript and would make the data more comparable with other studies. Author: We fully agree that performing more detailed analyses on interpolation and timeseries analyses would be possible and indeed relevant here. We decided to limit data interpretation here as we want to do this in collaboration with other researchers and in follow up studies. The argument being that interpolation and time series analyses could be an entire study in itself. We would therefore inadvertently end up excluding relevant methods should we choose different ones, as suggested here. We therefore, for the sake of simplicity chose to present the most conservative annual estimate, which was multiplying the average flux with 365 days. We truly believe that expanding collaborative potential with this open dataset will increase the impact beyond what could be done in one manuscript provided the extensive nature of the dataset.

l. 247/Figure 6: This figure should be moved to the appendix together with the following text in l. 253-261. As a modeler (which seems to be the main target group here), I might not be super interested in how exactly you processed the data, but want to see the clean data presented. Author: we have tried to reformulate so it appears already in the introduction that the data set is made available for both experimentalists and modelers, without favouring one group over the other. For the sake of length we agree that this detailed figure is better suited as a supplementary figure.

l. 263: Instead of "Ebullition, e.g. mass flow of CH4" I suggest "Methane ebullition flux" Author: done

l. 268 and 269: replace "enclosure-1" by "per enclosure" and clarify if you refer to a time or space component here – do you refer to 5 min? Please clarify. Author: done

l. 270: Equation 1 results in units nmol * 1e-6 * m-2, so I think 1e-6 should be removed? Author: done

l. 275: It is unclear how often ebullition was detected from the ditch, so the frequency remains unknown. Would it not be adequate to extrapolate ebullition fluxes throughout the day based on the measurements? Or is that what was done? Please clarify and elaborate. Author: we have now added in lines 339-346 the frequency of detected ebullition events and how this was used to extrapolate to an annual estimate. First, ebullitions represent the flux over a 5 minute flux measurement time as it is the accumulated $CH_4$ that counts. Ebullitions were detected in 334 out of a total of 1728 flux measurements from the ditch, e.g. 19.3% of the time. At the same time it can be assumed that the diffusive fluxes remain, so they are present 100% of the time. So to extrapolate to an annual estimate we estimated the number of 5 minute enclosures in the duration of the study period (360 days) where ebullitions were detected (19.3%). This gives a number of 5 min enclosures of 20049 for 360 days. This is multiplied with the average ebullition flux and summarized to g $CH_4$ m$^{-2}$.

l. 278-279: suggest to remove sentence Author: done

l. 279-284: I don't understand this, please clarify. I'm also not sure whether upscaling is the right term here or if you refer to extrapolation? Does this mean that there were 2 actual out of 5 potentially measured ebullition events? And therefore, for 3 out of 5 measurements, the ebullition flux was zero? If so, those should be considered too. Were there any measurements where you were able to determine both diffusive and ebullition fluxes? Was the concentration burst only observed for CH4 or also CO2? Authors: We appreciate this comment and after having reviewed the text we do realize it was actually presented wrong. $CO_2$ did not show ebullition behaviour as $CH_4$. We have changed the text (line 338-348) and only explained how we extrapolate to an annual estimate considering the frequency of ebullitions to diffusive fluxes:

"Out of a total of 1728 flux measurements from the ditch (collar 10), 334 were classified as ebullitions indicating that ebullition was erratic which is in line with studies of ebullition of fluxes from ponds (Wik et al. 2016; Sø et al. 2023). Hence, it can be assumed that ebullition occurred around 19.3% of the time during the measurement period (360 days). Furthermore, the ebullition flux is calculated as the accumulated CH4 in the chamber headspace during the entire flux measurement, e.g. 5 minutes here (Sø et al. 2023), and the calculated ebullition flux in the data set is therefore representative of 5 minute enclosure and not per second. To extrapolate to an annual estimate the number of 5 minute enclosures in 19% of 360 days is therefore estimated (N=20049 5-min 360 days$^{-1}$), multiplied with the average ebullition flux (nmol CH4 m-2 5 min-1).

Ebullitions could also be caused by mechanical disturbance of the chamber landing on the collar. Ebullition fluxes were discarded if the sudden increase in CH4 headspace concentration (Fig. S4) occurred 30 seconds after recorded chamber closure as this indicated bubbles released by chamber deployment on top of the collar."

Figure 7: Again, I think this should be placed in the appendix. Author: Agree. Figure 7 is now Figure S3

Data presentation:

l. 300 and 312: I don't think wind climate is a common term. Wind regime could be used for example. Author: changed to wind regime

l. 301: replace "max" with "maximum" Author: done

l. 301-303: A pattern is not super clear in Figure 8, so maybe best to remove this sentence? Author: done

l. 307-308: Sentence does not seem very meaningful, so it could be removed. Author: done

l. 311/Figure 8: In panel A, how can the maximum speed be lower than the mean? Also, error bars for the mean should be inserted. In panel B, reconsider the legend of the color palette. Only blue colors are visible in the wind rose. Authors: the gust wind speed is shown on the right y-axis. However, we have decided to remove panel A, as it is less relevant than the wind rose diagram.

l. 323/Figures 9-12: Consider showing the times of GHG measurements with shaded areas

spanning the y axis instead of lines on the x axis. Could the x axis ticks be improved to show each first of the month for example? Same comments apply to the following timeseries figures. Author: done. For figure 11 the WTD measurements corresponded to the measurement period of GHG fluxes. We have improved the ticks for the figures.

l. 327: Sentence is very vague. Seasonal variation in temperature is typical, not only in Denmark. One could either elaborate more on this, or remove the sentence. Author: done

l. 329-331: rephrase to "Monthly ranges of air temperatures (Tab. 2) show >20°C variation between minimum and maximum, except for February, pointing towards large diurnal variations." Author: done

l. 333-335/Table 2: Usually, the mean alongside the standard deviation would be given or standard error, for example. Please add that. Author: adding the standard deviation/SE is in our opinion for this specific table not more informative than having the max and min values of the shown variables. I would assume the reviewer wants the SDEV to provide a measure of variability around the mean. We think that having the max/min values achieves this in a more transparent way. fWe therefore suggest to keep the current design.

l. 339/Figure 10: What is the message of this figure? The soil temperatures would be expected to be very similar along the transect, unless there are clear differences between the locations. It is a bit unclear whether any of the sensors could for example be below the groundwater level at any point. One could show one figure instead that shows the overall seasonal cycle of temperature for all collars combined. Also, the numbers of collars are not consistent with those given in Table 1 for soil temperature measurements, collars 1 and 18 are missing. Is there a reason for that? Add legends for the line colors in the plots. The different blue colors in panels A-E are difficult to distinguish, please choose others. Author: The point was to show they are similar and yes, this is to be expected. However, we agree that this figure may be excessive and we have merged the soil temperature data in to one graph and added it to figure 5 as panel B.

l. 347-349: Is this information relevant for this study? Author: text has been modified to make it less repetitive.

l. 356/Figure 11: If the overall water table depth for the site is to be presented, then the figure does not necessarily help much, since the elevation of each collar is different. Figure 4 does a better job. Maybe the authors could clarify the main message of this figure. Is it relevant or would it fit better in the appendix? Also, it would be worth to insert a horizontal line at depth 0. Author: this figure shows the water table depth relative to the surface and not the hydraulic head. We find it relevant to present the finer details in the temporal variability of WTD and not the overall variation as shown in figure 4. So we would like to keep it in the main manuscript.

l. 394: Here and throughout the manuscript, mean fluxes should always be accompanied by an error estimate, such as standard deviation. Authors: standard error values have now been added

l. 396: I think there may be little spatial variation until the start of summer, but after that, most sites take up methane, while some emit. Can the moisture or water level be considered as drivers for methane at this site then? Author: we have added the sentence "The low spatial and

similar temporal variation between collars indicate the both hydrological indicators of SWC and WTD are poor predictors of $CH_4$ fluxes at this site."

l. 402: maybe microbes played a role here? Author: yes, evidently, microbes are the main producers of $CO_2$ in this soil, since the majority of plant derived $CO_2$ is avoided, but the microbial activity is only moderated by WTD and the spatial variability in WTD does not seem to be able to predict spatial patterns of $CO_2$.

l. 404-406: is there any information available about nutrient content in soils? Author: as per the request of R3 we have added a section in the Data presentation data on soil and groundwater chemistry. We refer to these specific comments below.

l. 407/Figure 14: This figure is a very nice visualization of the study. Consider replacing Figure 4 with Figure 14 to avoid repetition. Author: thank you. We have removed figure 4 and 14 and it is now Figure 3.

l. 426-428 and 430-431: a bit awkwardly phrased sentences, consider to rephrase slightly for better understanding. Author: we have tried to improve the language of this text in lines 481-491.

l. 436: This is the first time in this section that results were discussed and compared with those from other studies. I definitely miss a bit more discussion in this manuscript. Please add references and discussion for the other gases as well. Author: We have deleted this sentence where the hotspots have been compared to a temperate forest and replaced the citation with a more relevant article from a temperate peatland (Anthony and Silver 2023). Again, we want to reiterate that this paper is not intended for discussion of mechanisms, but to present that data. We believe that we are limited by space to fully discuss in depth the hot moments aspects of the $N_2O$, but have provided tantalizing evidence that this is a major driver here.

l. 439/Figure 16: Small text on right side of the figure to indicate months is barely visible. Author: We have updated the figure to make the text on the right side bigger

l. 443: Maybe better to state "we observed a clear diurnal cycle" or similar. Author: done

l. 444-446: Split up or rephrase sentence to avoid long chain of information. Author: done

l. 450/Figure 17: What is shown here? Are the fluxes averaged across collars and then summed? Is WTD also averaged across collars? If they are identical for subplots A-C, then there is no need to show it 3 times. Please revise and clarify in the caption. Authors: It is now stated that soil collar fluxes were average and summed to monthly site sums. In the caption it is stated that the WTD is also site average. We prefer to keep the WTD curve for panels A-C to more clearly relate the WTD and temperature dynamics to all gases.

l. 456: The relation could be shown explicitly in a correlation matrix or plot. In Figure 17, the relation is not clear. Author: we prefer not to perform correlation analyses as we think this necessitates a reframing of the entire manuscript away from the data description. We have modified the wording of this sentence (line 552-556).

l. 459 and following: Please make sure there is no repetition of this information. It might make

sense to restructure or comprise section 3.6 a bit to avoid repetition. Author: In line 557-565 we have now explicitly mentioned $CO_2$ and $N_2O$ and therefore these sentences regarding $CH_4$ are descriptive of its temporal variability. We prefer to keep it as, in our opinion, it does not repeat information.

l. 474/Figure 18: Is it possible to partition the diffusive and ebullition CH4 fluxes (considering your methodology) to see whether the higher CH4 fluxes result from ebullition? Please add horizontal lines at zero. Author: In the data file in the repository the diffusive and ebullition fluxes are separated and hence it is possible to partition the contribution. This is explicitly done in the monthly and annual sums, but we here we prefer to show the total CH4 emission per day. Horizontal lines have been added.

l. 480: Please be more precise about the frequency/occurrence of ebullition at the ditch. How often did it occur? What was the magnitude of the fluxes? And how does this compare to diffusion? Author: we have added the frequency of ebullition measurements and its average contribution to CH4 emissions.

l. 502: Worth mentioning that ebullition fluxes are very sporadic and that there is no clear temporal pattern to be expected. Author: a sentence has been added in line 607-608 including two relevant references.

l. 511-512: The figure reference "(Fig. 19D)" should probably follow after "zero", and reference to Figure 18C could be added at the end of the sentence. Author: done

l. 514/Figure 20: Please add horizontal line for zero or add shaded area or similar. Done?

l. 519: Is the proportionality based on visual inspection of Figure 20? Author: We have deleted the wording proportionality. Yes, it was based on visual interpretation.

l. 546-547: I think a reference is needed here when referring to NO3- measurements which are not presented in this study. Author: as mentioned earlier. A new section in Data presentation is added showing $NO_3^-$ data in groundwater as per request of R3.

l. 554-555: Might be worth to elaborate on this. Author: We believe we have addressed this already in lines 571-586.

l. 559/Table 3: Footnote about GWP can be removed since the info is given in the table caption. Why is the 20-year GWP given only for the ditch? What are the numbers in parenthesis for the ditch GWP for CO2 and N2O? What are the numbers in squared brackets? It could be worth to add further estimates based on different approaches here, as indicated in earlier comments. Consider using <1 or similar for peat soil CH4 GWP since it is not equal to 0. Could the fluxes also be given as area-weighted (in addition to per-m2) fluxes if the area of peat soil and the ditch is known? Author: The numbers in brackets were for the GWP20, but we decided to remove all mentioning of global warming in the table since it is less relevant for this manuscript and because we lack the net ecosystem exchange of $CO_2$. Also, we would like to refrain from providing an area upscaled GHG estimate since the fluxes here only represent a smaller area of the entire peatland at Vejrumbro. Regarding the other interpolated annual estimates: as per our argument earlier we here present the most conservative estimate possible.

Conclusion:

Please list limitations of the study and answer the questions: Would the budgets in a different year likely be the same? What effects does the disturbance history of the site have on the measured parameters? I. e. is the data representative in time and among temperate fens? Author: We have added a sentence mentioning that this dataset is only to be viewed as being representative of the period as other climatic and hydrological conditions in other years will lead to different fluxes (lines 650-652).

We have also added a summary paragraph on the effect and limitations of the disturbances we made at the site (lines 653-661).

The text in lines 662-664 states the comparison to other studies, which has also been outlined in the text (lines 472-480).

Supplement:

It is unclear which plot belongs to which collar. Please indicate. Also, why was collar 10/the ditch excluded? Author: collar 10 was excluded here as the groundwater data could not be represented as depth below terrain. Hence a graphical streamlining with the soil plots was not possible. Also, the timeseries of ditch GHG emissions is included now as figure 18. However, the ditch water temperature has not been presented, but is included in the data repository file "VB groundwater depth Figure 6.xlsx". We judge this as a minor issue and would like to keep the supplementary figures of collars 1-27 (excluding 10) as it is. We have detailed which plot belongs to which collar and explicitly stated that it is only for soil collars.

**Reviewer #3 Ko van Huissteden**

This data paper is a useful contribution to the study of soil greenhouse gas fluxes in peatlands, because it presents data from a novel system of automatic chamber greenhouse gas measurements covering a transect with a large number of collars, a relatively high data frequency and a full year coverage. However, I have two serious objections to this paper that should be addressed properly before final publication. Therefore I recommend publication, but with major revisions. My main objections are:

1. A lack of ancillary soil and vegetation data. There is practically no data on soil and vegetation included, except for a short mentioning of peat soil. Neither it is clear which of the cited references gives adequate information on soil conditions of the transect. If these data are to be be used by other researchers on ecosystem greenhouse gas fluxes, one would at least expect a basic description of a soil profiles at the site, or a borehole transect, and some basic soil and water chemistry data.

Author: we have added new sections 2.1.2 Peat soil characteristics (lines 143-161) and description of water sampling in lines 162-175 2.1.3 Groundwater water sampling and chemical analysis and in section 3.6 Groundwater chemistry (lines 414-446) (+new figure 8). In these new sections we present the methodology of soil and groundwater sampling and describe magnitude and trends. This data has now also been included it in the published dataset. Since vegetation was removed and we did not excavate roots we do not have vegetation data.

2. The measurement procedure, that entails removal of vegetation. The procedure of removal of vegetation has been common in the past but is increasingly abolished because of the intense intertwining of vegetation, microbial community and soil processes that generate greenhouse gas fluxes. Removal of vegetation (including the application of herbicide in this case) is a large disturbance of this system, with questionable results. It introduces artefacts that are poorly quantified for $CO_2$ fluxes since labile carbon pools in the soil are affected.

In the case of $CH_4$ fluxes, the main supply of labile carbon for methanogens is reduced, and the main transport pathway of $CH_4$ from soil to atmosphere (by plant aerenchyma) is destroyed. Therefore it likely leads to much lower fluxes of $CH_4$ compared to those in an undisturbed system, resulting in data that cannot be compared to those of other sites – if not simply flawed. For details, see the comments below.

The authors should state clearly in the abstract that vegetation removal has been applied. Furthermore, they should discuss properly what effects this may have had on the fluxes that they have measured.

Even if this is a data paper only, reflection on the complexity of the system that you have measured, and on the effects of your measurements on that system, is necessary.

Author: Thank you for this comment and the concerns you raise are shared with R2. We have included more text on the potential effects of glyphosate addition and vegetation removal in lines 120-139. We fully acknowledge that plant removal is suboptimal, but with the current chamber setup it was a necessity to be able to measure the fluxes. The limitations of plant removal have been mentioned now and it is stated in the conclusion that these data do not represent net ecosystem exchange data and that especially $CH_4$ fluxes may be biased because of the role of the plants. Visual inspection at the site confirmed that there is considerable lateral root growth underneath the soil collars from adjacent vegetation as we only removed vegetation

from a 40x40 cm area around each collar (now detailed in description of plant removal). Hence, it is likely that plant derived C and N substrates for methanogens and N cycling microbes could be present in the soil under the collars. However, we did not excavate roots during the study to avoid excessive disturbance.

You are right that the artefacts that are introduced are unknown and difficult to quantify. Since we did not have an undisturbed control we cannot quantify the disturbance effect, but only discuss potential biases it might have introduced. We think also it is fair to say that all chamber based systems have biases and are essentially impacting the system they measure on. Only eddy covariance can be considered truly non-invasive.

Minor comments concern mostly the quality of figures and their captions, and questions on the operation of the automatic chamber system.

**Detailed comments on the paper.**

Section 2.1 Site description: This is disappointingly incomplete. Not any information is given on the soil profile and it lateral variation along the transect, while this could have been checked with a few hand augerings. What is the peat stratigraphy, are there any sand or clay layers in between or on top of the peat? What is the peat type, its decomposition grade, loss on ignition? Any information about soil water chemistry, for instance the presence of anaerobic electron acceptors that influence the redox potential and methanogenesis? What is the variation of the vegetation along the transect? *Juncus effussus* and most grasses differ strongly in the characteristics of their root system and methane transport characteristics. All this is information that any user of your data would want to know.

Author: Thank you for this helpful comment. We realize now that this important site information was missing and we have now added these descriptive data under the Site description section of the paper. See also comment above regarding this.

Line 94 – 101, caption Fig. 3: What are the instruments in the lower right corner of the figure? Author: they are automated chambers measuring light and dark fluxes of $CO_2$ and $CH_4$ on Juncus and reed canary grass plots. These instruments belong to another research project.

Figure 4: The figure is not very informative (except on the surface topography and placement of the collars) and the caption is confusing. The vertical profile has two colours, brown and dark grey, which suggests some sort of stratigraphy. However, the brown colour is marked as 'transect surface', but apparently it indicates the soil above the minimum water table depth. The lower depth of the peat is not indicated. Information on the peat properties and its variability (e.g. the presence of clay/sand layers) is lacking. This would be very useful information for users of the data. Author: we agree that the use of brown color and blue shaded area can be misleading. Now that we have included peat soil characteristics prior to this figure we only show the transect surface, with the collars, piezometer locations indicated as well as the mean, max and min WTD. As per suggestion from R2 the original Fig. 14 and Fig. 4 and is now Fig. 3 was merged and Fig. 14 was deleted. We would like to maintain this overview figure of the transect as we believe it is informative for the reader to understand how the transect was configured and instrumented.

Line 170 – 173:

This procedure of killing vegetation by harvesting, and application of a herbicide, attempts to reduce the effects of vegetation respiration and to measure the 'true' or 'net' soil GHG flux.

However, it introduces other artefacts that are poorly quantified, in particular for the $CH_4$ fluxes. For measuring of $CO_2$ fluxes from the soil it often has been done with the purpose of reducing $CO_2$ respiration/uptake by plants. Because of the artefacts it introduces, alternative approaches have been developed that leave vegetation intact and separate soil and vegetation components of the flux by modelling (e.g. Boonman et al., 2024). For the $CH_4$ fluxes it may have resulted in serious underestimation of the fluxes. Author: as mentioned below the chamber design did not allow vegetation to be present due to the small volume of the chamber relative to the high (up to 100 cm) vegetation at the site. We are aware of the underestimation of $CH_4$ fluxes we may have caused from the vegetation removal and address this bias throughout the text using some of the references suggested below.

The statement that the fluxes after removal of the vegetation represent the 'net' soil greenhouse gas flux is invalid without specification what is actually meant by 'net flux', in particular when it is not explained which soil carbon pools are assumed to contribute to to this net flux. It may at best approach the soil $CO_2$ flux (with an unknown error or bias) and likely severely underestimates the $CH_4$ flux. In general, $CO_2$ from the decomposition of recently produced labile carbon, and that from older soil carbon (e.g. the peat carbon pool) is difficult to separate in surface flux measurements. Vegetation removal does a poor job in that, because most the root mass often remains behind, will be active, and also affects the microbial population. Author: we have in lines 136-137 now added that the net soil GHG flux we measure consists of heterotrophic respiration and autotrophic respiration of live roots from adjacent plants that were present below the collars.

Besides these caveats, I also wonder if vegetation removal is a specific requirement of this automatic chamber system. Does vegetation hamper a leak-free placement of the chamber on the collars with this system?

Author: as now detailed in lines 107-119 it was necessary to remove tall vegetation due to the limitations of the chamber system. We are fully aware that this is suboptimal and we address this now in the paper. However, all chamber systems are flawed and there will always be a trade-off. In our specific case we then focused on the relation of spatiotemporal variation of net soil fluxes to hydrological and thermal drivers.

A motivation why this procedure has been applied, and a discussion of the caveats listed below is necessary. I suggest to do this in a separate Discussion section. Also, the vegetation removal procedure itself should be mentioned clearly in the abstract, for potential users to judge wether the data are suitable for use. Author: Agree that more discussion on the impact of vegetation removal is needed, which was also raised by R2. Therefore, we have added text throughout the manuscript discussing this. However, we are reluctant to dedicate an entire section of the discussion on this as we are limited in quantifying the artefacts/biases of the vegetation removal on the fluxes, as we did not have an undisturbed control plot. We have included a section 2.1.1 Site preparation and disturbance (lines 106-139) under 2.1 Site description to present and discuss these concerns early in the manuscript as a prerequisite for the reader to interpret data presented later in the manuscript.

Drawbacks of vegetation removal:

1. The soil greenhouse gas flux in an undisturbed ecosystem is the sum of all peat and other organic matter decomposition. The 'other' being various forms of recently produced, usually labile carbon, produced by the vegetation root system and litter decomposition. Removal or

decrease of one carbon pool may strongly affect the measured fluxes, in particular when it is not known quantitatively what has been removed. Furthermore, labile carbon interacts also with stabile carbon decomposition via the priming effect. This may enhance stable carbon decomposition (e.g. peat decomposition) in the presence of labile carbon. Therefore, it should be specified which carbon pools are considered to be included in the flux measurements (peat, older humic matter, recently produced organic matter, labile or stable?), and what effects the measurement procedure has on $CO_2$ emission from these pools. If actual data collection, e.g. root mass, is not available, the authors could at least consult literature from other sites on that.

Authors: we have written in lines 109-112 the following: "The strategy was therefore to focus on measuring net soil GHG fluxes, where we assume the production of GHG are derived from heterotrophic respiration of older peat C/N, root exudated C/N from adjacent plants, dissolved N in groundwater and belowground autotrophic respiration from roots inhabiting the peat below the collars. We did not sample the live root mass in the peat below the collars and we are unsure how to include root data from other sites as this will also introduce uncertainty related to site specific differences in productivity and plant species. We hope that with this addition we can clarify to the reader what the main assumptions were and that we are aware of the potential caveats that need to be dealt with if using these data.

2. Glyfosate is known to affect soil faunal and soil microbial respiration (e.g. Nguyen et al., 2016). The application of this herbicide will have influenced the measured fluxes to an unknown extent. Authors: We have added a reference (Padilla & Selim 2020) to a paper addressing the half-life of glyphosate in soils. Overall, it shows that the half-life is short and we consider the direct effect of glyphosate to be small. Thus, we have added additional discussion on the potential effects on glyphosate on microbial activity and mention that in our case we cannot fully quantify the effect. Although, we only applied it once and not continuously and considering a short half-life in soils the direct effect is assumed to be small in our case.

3. As the need for very frequent removal of living vegetation during the experiment testifies, the root system in the soil remained active, producing labile carbon and adding a vegetation and labile carbon respiration component to the fluxes. Therefore, vegetation removal still does not remove vegetation effects. Author: agree. We have added that the fluxes also represent an autotrophic belowground component.

4. Since detection of $CH_4$ emissions is included, you are removing one of the main transport mechanisms of $CH_4$ from soil to atmosphere: the transport via plant aerenchyma (e.g. Vroom et al., 2022). Moreover, the main source of carbon for methanogens is labile carbon compounds produced by plant roots. The low $CH_4$ emissions therefore may be flawed and not represent normal ecosystem or soil $CH_4$ fluxes. On peat soils with approximately similar water table variation and vegetation, significant positive $CH_4$ fluxes were measured with manual and automated chambers (Hendriks et al., 2007; Lippmann et al., 2023). Author: Thank you for the relevant references to discuss the caveats regarding the $CH_4$ fluxes and vegetation. We have now specifically mentioned in lines 459-461 that we likely have limited $CH_4$ emissions due to removal of aboveground vegetation. This has also been restated in the conclusion. However, due to the presence of roots from adjacent vegetation it is likely the substrate supply remains and also the WTD was rather deep (20-40 cm) in the growing season, which would not necessarily favour high $CH_4$ production rates in the top soil in addition with the presence of major electron acceptors. We have added these discussion points to the text as well as the Vroom et al. 2022 paper.

Furthermore, you say that you removed vegetation with a minimum of 7 days. What was the (probably higher) vegetation removal frequency in the spring and summer period? This is highly important given the rapid vegetation regrowth in that part of the year. Author: a 7 day interval was adequate to avoid regrowth. Furthermore, we never observed net $CO_2$ uptake in the growing season indicating that the vegetation removal was effective.

Line 179: How does wind speed affect the operation of the system? How reliable is it at higher wind speeds? Author: there will always be uncertainties under high wind speeds. However, provided that the chamber was equipped with a vent the adverse impact of wind was lowered. We did not observe more variable headspace gas concentration behaviour under strong winds, indicating that the vent was working properly. However, the SkyLine2D system was robust to wind movement and continued operation in all the wind regimes encountered at the site.

Line 180: How certain can you be that rapid vegetation growth near the collar does not affect the airtight connection of chamber and collar? For instance, leaks may result from high grass getting between the chamber and the gasket during windy conditions. Author: because vegetation was never allowed to grow above the collar. Combined with vegetation removal outside of the collar there was no opportunity for vegetation to get stuck in between the chamber seal and the collar.

Line 182: If a fan was not installed in the chamber, what is the air flow provided by the main pump, and is it sufficient to flush the chamber? Author: yes, as shown now in Figure S2 the mixing of the chamber was sufficient with the main pump. When fluxes became lower the headspace concentration also became more variable as is also seen with manual chambers.

Figure 5: Can the upper photos be made sharper or larger, providing more detail on the chamber construction? Eventually, provide them in a supplement. Author: Figure 5 has been moved to supplementary and is now called Figure S2. We have updated the figure with higher resolution photographs of panels A – C.

Line 232: Check the sentence starting with "If the relative SE...", the part "than 100%" appears to be misplaced. Author: There was a typo and the text (line 304-306) has been corrected.

Line 238: During the measurement time the temperature in the transparent chamber can rise in a matter of a minute in sunny weather. Is there no temperature sensor inside the chamber to detect this effect and correct for it? Author: there was no temperature sensor inside the chamber. It is true that the headspace temperature can heat up, which would mainly impact the conversion of ppm to moles. However, the increase in temperature increases in theory the partial pressure of headspace gases, but the vent ensures a pressure buffer (section 6 in Clough et al 2020; section 2.2.3 in Maier et al. 2022). The temperature effect would be highest on non-linear behaviour of gases, but the use of HM accommodates for the non-linearity and estimates the slope at time zero. If the headspace concentrations behave linearly it must also be assumed that the temperature effect is negligible and hence do not impact flux estimates. There is also a theoretical possibility of a warmer headspace heats up the soil below the chamber, but given that flux measurement duration was only 5 minutes it is unlikely that the soil would be heated up. In conclusion, despite we did not measure the temperature we do not see a need correcting for increased chamber headspace temperature as 1) the chamber was vented alleviating temperature induced pressure changes and 2) the short measurement duration prevented any temperature increase to extend into the soil and change biological processes.

Caption Fig. 7. Good point, but how sensitive is the system to bubble fluxes induced by the chamber lowering? Was the collar anchored somehow in the subsoil of the ditch to prevent disturbance? Author: It has been added in the text that the collar was glued on top of a longer perforated tube that was anchored in the bottom of the ditch. In lines 346-348 we describe that if bubbles were monitored 30 seconds after chamber closure it would be an indication of mechanical disturbance and this flux would be discarded.

Line 383: variability in soil water content. Again, it is disappointing that so few soil data is included. Could there be differences in water seepage from higher ground (which is to be expected given the topography) or does soil cracking in dry periods occur, influencing the SWC? Author: we have now included peat soil characteristics. The peat characteristics are relatively homogeneous across the transect and we never observed soil cracking at this site.

Line 398 – 402. Again, the large spatial variability should be no surprise. Unfortunately, any information on soil variability is missing. For instance, I would have expected information on the soil carbon content, which is an important predicting variable in $CO_2$ fluxes from soils. Although the above-ground vegetation is removed, there is still root mass present that produces labile carbon; root density also adds to the spatial variability. This kind of data would be very useful for other users of the data. Author: according to the peat soil data presented now in table 1 indicates little spatial variation in TC and TN. We do not have data on labile carbon pools from roots, but we agree this would have been very useful.

Line 424. Interesting to see these bursts. As mentioned above, this could also be an artefact of your methodology. It excludes plant emissions, which is usually a major $CH_4$ emission pathway (Vroom et al., 2002). By artificially removing the plant flux, a buildup of $CH_4$ concentration in the soil could induce a burst-like emission pattern. Author: yes, the burst-like emission patterns are most likely due to $CH_4$ concentration buildup that is not "vented" by the plants. This mechanism has been added to the text.

Figure 16 and Figure 19. This figure is difficult to understand, information in the caption is ambiguous. Author: agree. We have clarified the caption for these figures. Note that they have been given new numbers, Fig. 16 → Fig. 11 and Fig. 19 → Fig. 14.

You have only 5 measurement points per day, but there are more observation points. The caption suggests that the points are based on a grouping of all soil collars together. Is this over one day, if so, which days? Or over an entire month, as the reference to the figure legend suggests? In that case I would have expected way more data points in the figure, unless it is a monthly average per collar. In short, be clear how your data have been grouped. This is also important for understanding sources of variation in the data. Grouping of all collars in one day introduces also spatial variation, next to temporal variation. Author: yes, you are right. We assigned each flux observation to the hour of the day (e.g. between 1 – 24) and averaged these per month. This is now written in the caption.

Next, the colour scale of the legend is not very distinctive by choosing only shades of blue and red. Better include other colours as well, which makes the data from different months more distinctive. Author: the colours for the different months were chosen to indicate that blue is colder and red is warmer. Initially, we did try different color schemes, but they did not work and just created more visual confusion in our opinion. To help the reader distinguish between the different months (e.g. colours) we added in small text the month abbreviation right of the LOESS

curves. If the manuscript is accepted for publication a high-resolution version of this figure will be available.

Line 443-444. The lack of diurnal variation for $CH_4$ may also have been caused by removing the vegetation. Plant fluxes of $CH_4$ tend to have diurnal variation, driven by solar radiation (Vroom et al., 2022). Author: we agree with this and have added further details and citations in lines 568-577.

Line 540. 'it cannot be ruled out that living roots inhabited the soil below the chambers'. You can be quite sure about that if you have to clip the vegetation frequently! Author: yes. We have included more reflections throughout the text on this and how it impacted fluxes.

Line 547-550. There could be other electron acceptors inhibiting methanogenesis, for instance $Fe^{3+}$, sulfate. Here again, some information on basic soil and water chemistry could have been helpful for users of the data. Author: we have now added groundwater chemistry data including major electron acceptors. We also added some discussion on how it may impact $CH_4$ fluxes in lines 469-471 and lines 672-674. Indeed the presence of both $SO_4^{2-}$ and $NO_3^-$ as well as Fe indicate that there are enough electron acceptors to prevent methanogenesis to happen.

Table 3. Mention the source of the GWP factors used here. Author: we have added this reference

Section 5 Conclusions: The causes for spatial variability of the GHG fluxes is unresolved – but that is not surprising given that any information on soil variability is lacking. Fig. 15 suggests that the spatial variability is larger than the temporal variability on these closely spaced collars, which would be an interesting conclusion. Author: we have expanded this sentence in line 646-649: " The cause for the spatial variability of GHG fluxes remains unresolved and do not clearly link directly to either WTD, soil temperature and soil/groundwater chemical parameters. Interestingly it appears that the temporal variability of GHG fluxes across the transect is lower than the spatial variation."

Line 578-580: The low $CH_4$ emission is attributed to low water table and a cold wet winter. However, the huge elephant in the room here is the potential effect of vegetation removal on the $CH_4$ fluxes detailed above. At other peat sites with similar water table and vegetation that I have measured myself, persistent positive summer $CH_4$ fluxes occurred (Hendriks et al., 2007; Lippmann et al., 2023). Neither, alternative explanations for the low emissions are considered, such as the presence of other anaerobic electron acceptors (e.g. sulfate reduction) that maintain a too high redox potential for methanogenesis? This is mentioned elsewhere in the article for $NO_3^-$, but not considered here. Author: we agree with this comment and have added to the conclusion (line 674-676): "However, it cannot be ruled out that the vegetation removal impeded $CH_4$ emissions, as we effectively restricted plant mediated $CH_4$ emissions. Therefore, caution should be taken when comparing the $CH_4$ flux data to other drained peatlands. Furthermore, electron acceptors present in groundwater suggest that redox potential may have been too high to sustain $CH_4$ production."

**Supplement.**

Missing: collar numbers at each graph. What do the ticks on the horizontal axis represent? First day of each month, midpoint? Day numbers would be more informative on the horizontal axis! Author: Collar numbers have been added to each graph. It has also been added in the description of this Figure S4 that the tick marks represent the 1st day of each month.

**Data.**

The data representation is largely correct. However, for greenhouse gas fluxes it would be useful include the standard error of the flux calculation method that is applied. This would allow data users to apply additional quality checks. Author: agree the standard error of the slope estimate (ppm s-1) has been added to each flux in the data file "VB GHG fluxes Figures 9 - 15.xlsx"

**References.**

Boonman, J., et al. (2024):Transparent automated CO2 flux chambers reveal spatial and temporal patterns of net carbon fluxes from managed peatlands. *Ecological Indicators*, *164*, 112121.

Hendriks et al., (2007): The full greenhouse gas balance of an abandoned peat meadow. Biogeosciences, 4(3), 411-424

Lippmann et al. (2023): Peatland-VU-NUCOM (PVN 1.0): using dynamic plant functional types to model peatland vegetation, $CH_4$, and $CO_2$ emissions. *Geoscientific Model Development*, *16*(22), 6773-6804.

Nguyen et al. (2016): Impact of glyphosate on soil microbial biomass and respiration: a meta-analysis. Soil Biology and Biochemistry, 92, 50-57.).

Vroom et al., (2022): Physiological processes affecting methane transport by wetland vegetation, Aquatic Botany 182:10354

Author: thank you for pointing our attention to these relevant references. We have used the Boonman et al to highlight the use of automated chambers in peatland research. Nguyen et al. to address the glyphosate issue and Vroonm et al. to frame the results in relation to plant mediated $CH_4$ fluxes

Author: References used in this reply

Brændholt, A., Steenberg Larsen, K., Ibrom, A., & Pilegaard, K. (2017). Overestimation of closed-chamber soil CO2 effluxes at low atmospheric turbulence. *Biogeosciences*, *14*(6), 1603–1616. https://doi.org/10.5194/bg-14-1603-2017

Padilla, J. T., & Selim, H. M. (2020). Environmental behavior of glyphosate in soils. *Advances in Agronomy*, *159*, 1–34. https://doi.org/10.1016/BS.AGRON.2019.07.005

Clough, T. J., Rochette, P., Thomas, S. M., Pihlatie, M., Christiansen, J. R., & Thorman, R. E. (2020). Global Research Alliance N 2 O chamber methodology guidelines: Design considerations. *Journal o  Environmental Quality*, jeq2.20117. https://doi.org/10.1002/jeq2.20117

Maier, M., Weber, T. K. D., Fiedler, J., Fuß, R., Glatzel, S., Huth, V., Jordan, S., Jurasinski, G., Kutzbach, L., Schäfer, K., Weymann, D., & Hagemann, U. (2022). Introduction of a guideline for measurements of greenhouse gas fluxes from soils using non-steady-state chambers. *Journal o  Plant Nutrition and Soil Science*, *185*(4), 447–461. https://doi.org/10.1002/jpln.202200199

---

## Referee Report (RR1)

Second review of:

**A full year of continuous net soil and ditch CO2, CH4, N2O fluxes, soil hydrology and meteorology for a drained fen in Denmark**
Annelie Skov Nielsen, Klaus Steenberg Larsen, Poul Erik Lærke, Andres Felipe Rodriguez, Johannes W. M. Pullens, Rasmus Jes Petersen, and Jesper Riis Christiansen

After my first review of this paper, I see that the authors have done an excellent job in improving the paper. Important ancillary information on the soil profile and water chemistry of the site has been added, which improves the usefulness of the greenhouse gas measurements considerably. The drawbacks of the measurement system and the procedure of vegetation removal have been discussed properly, albeit that the effects of vegetation removal still cannot be quantified due to the lack of a control experiment, as is admitted by the authors. However, such a quantification was not the goal of their measurements. Representing greenhouse gas measurement data from a novel chamber measurement system is in itself a very useful contribution to the research field.

A few smaller matters remain, which could be tackled by minor revisions.

- Effect of vegetation removal on $CH_4$ emission (reply by authors, page 18, point 4; revised text, line 459 – 465). Here, the authors first state that the effect of vegetation removal on the $CH_4$ flux might not be very strong because of root growth from outside into the collars, and second, the lack of labile carbon supply would not have a large effect during periods of lower water table. However, the labile carbon supply to methanogens is hampered mainly by green vegetation removal, root mass effects are secondary. Photosynthesis is the actual source of labile carbon products, which may be transferred to the soil via the roots in a matter of hours to a couple of days. This has been proved by carbon labeling experiments, see e.g. King and Reeburgh, 2002 (King, J. Y., & Reeburgh, W. S. (2002). A pulse-labeling experiment to determine the contribution of recent plant photosynthates to net methane emission in arctic wet sedge tundra. Soil Biology and Biochemistry, 34(2), 173-180). So, by removal of green vegetation inside the collars you will inevitably cut off an important labile carbon source. Neither does the argument of lower water table hold. Roots of wetland plants such as sedges and *Juncus* can penetrate quite deeply and still add labile carbon to completely saturated soil, fuelling methanogenesis. In addition roots and stems will continue to transport some of the $CH_4$ towards the atmosphere, bypassing oxidation within the unsaturated topsoil. Therefore, the effect of vegetation removal on the fluxes will be considerable, and may occur even in drier periods with lower water tables.

- 7-day period of green vegetation removal (reply by authors, page 19, top paragraph). "Furthermore, we never observed net $CO_2$ uptake in the growing season indicating that the vegetation removal was effective." This is flawed reasoning. To my experience there can be a significant regrowth of vegetation in a few days during the growing season, resulting in a measurable $CO_2$ uptake that reduces the measured net flux. The fact that you never observed net $CO_2$ uptake, does not mean that your $CO_2$ fluxes are not influenced by photosynthesis of the small amount of leaf and shoot regrowth that may occur within seven days. This photosynthesis might not be able to overcome the soil $CO_2$ flux, but still will result in a reduction of the measured total flux. This should be mentioned in the text.

- Temperature measurements inside the chamber (reply by authors, last reply on page 19). I agree with the authors that in this case significant effects of a temperature rise in the chambers are unlikely. However, the conversion from ppm to moles will still be improved by adding a temperature sensor.

---

## Author Response (AR2)

Second review of:

**A full year of continuous net soil and ditch CO2, CH4, N2O fluxes, soil hydrology and meteorology for a drained fen in Denmark**

Annelie Skov Nielsen, Klaus Steenberg Larsen, Poul Erik Lærke, Andres Felipe Rodriguez, Johannes W. M. Pullens, Rasmus Jes Petersen, and Jesper Riis Christiansen

Author: We would once again extend our gratitude to all three reviewers for taking your time to further improve the manuscript with your insightful and constructive comments and suggestions. We have addressed each comment below and the second review have resulted in substantial changes to the structure of the manuscript. As per suggestion of Judith the Materials and Methods section has been rearranged to be less redundant. All time series figures have been edited to streamline time format and the conclusion has been restructured. Furthermore, where relevant and suggested we have made changes to the text.

**Report 1 by Ko van Huissteden**
After my first review of this paper, I see that the authors have done an excellent job in improving the paper. Important ancillary information on the soil profile and water chemistry of the site has been added, which improves the usefulness of the greenhouse gas measurements considerably. The drawbacks of the measurement system and the procedure of vegetation removal have been discussed properly, albeit that the effects of vegetation removal still cannot be quantified due to the lack of a control experiment, as is admitted by the authors. However, such a quantification was not the goal of their measurements. Representing greenhouse gas measurement data from a novel chamber measurement system is in itself a very useful contribution to the research field.
Author: Thank you for this comment. We appreciate that our efforts to improve the manuscript are acknowledged.

A few smaller matters remain, which could be tackled by minor revisions.

- Effect of vegetation removal on $CH_4$ emission (reply by authors, page 18, point 4; revised text, line 459 – 465). Here, the authors first state that the effect of vegetation removal on the $CH_4$ flux might not be very strong because of root growth from outside into the collars, and second, the lack of labile carbon supply would not have a large effect during periods of lower water table. However, the labile carbon supply to methanogens is hampered mainly by green vegetation removal, root mass effects are secondary. Photosynthesis is the actual source of labile carbon products, which may be transferred to the soil via the roots in a matter of hours to a couple of days. This has been proved by carbon labeling experiments, see e.g. King and Reeburgh, 2002 (King, J. Y., & Reeburgh, W. S. (2002). A pulse-labeling experiment to determine the contribution of recent plant photosynthates to net methane emission in arctic wet sedge tundra. Soil Biology and Biochemistry, 34(2), 173-180).
So, by removal of green vegetation inside the collars you will inevitably cut off an important labile carbon source. Neither does the argument of lower water table hold. Roots of wetland plants such as sedges and *Juncus* can penetrate quite deeply and still add labile carbon to completely saturated soil, fuelling methanogenesis. In addition roots and stems will continue to transport some of the $CH_4$ towards the atmosphere, bypassing oxidation within the unsaturated topsoil. Therefore, the effect of vegetation removal on the fluxes will be considerable, and may occur even in drier periods with lower water tables.
Author: Based on this comment we realize our argumentation may have been too simplistic and have modified this text to reflect the above mentioned processes (lines 458-461): "However, as we excluded plants from the collars we might have decreased the net emission of CH4 directly by restricting gas transport in aerenchyma from deep peat layers potentially sustaining net CH4

emission even though the observed growing season WTD was 20-40 cm (Askaer et al. 2011; Vroom et al. 2022) and indirectly by potentially reducing plant carbon supply to methanogens." Furthermore, we have deleted the text: "However, visible inspection at the site confirmed lateral root growth from vegetation adjacent to the collar. This could indicate that plant derived C and N was still available for microbes underneath the collars, but the impact on gas transport is uncertain. However, we did not excavate roots during the study to avoid excessive disturbance. Furthermore, considering that the WTD in the growing season was mostly 20-40 cm below terrain the potential for $CH_4$ production in the topsoil would limited (Koch et al. 2023)." Finally, we have modified the text in lines 461-463: "The lack of consistent hot moments of $CH_4$ emissions and low cumulative emissions during periods of shallow WTD in the growing season (Fig. 6A-F), potentially conducive $CH_4$ production, could indicate that redox potential is elevated due to presence of other electron acceptors."

- 7-day period of green vegetation removal (reply by authors, page 19, top paragraph). "Furthermore, we never observed net $CO_2$ uptake in the growing season indicating that the vegetation removal was effective." This is flawed reasoning. To my experience there can be a significant regrowth of vegetation in a few days during the growing season, resulting in a measurable $CO_2$ uptake that reduces the measured net flux. The fact that you never observed net $CO_2$ uptake, does not mean that your $CO_2$ fluxes are not influenced by photosynthesis of the small amount of leaf and shoot regrowth that may occur within seven days. This photosynthesis might not be able to overcome the soil $CO_2$ flux, but still will result in a reduction of the measured total flux. This should be mentioned in the text.

Author: We agree that the net flux we measure may be positive even though there is photosynthesis. However, if there indeed was a significant regrowth of plants that would photosynthesize and hence decrease net CO2 effluxes, we should expect to see a systematic decrease of net CO2 effluxes over 7 day periods during the growing season. This was never observed. We do not have photos to document reemergence or other data to estimate plant abundance, so our conclusions here are based on the visual inspections done at the time of removal and the overall knowledge of the flux magnitudes and temporal variability in the growing season. We have modified the text in lines 126-131 to reflect the above: "Regrowth inside collars was manually removed at least weekly, minimizing photosynthetic $CO_2$ uptake. While regrowth abundance was not measured, stable net $CO_2$ efflux between removals suggests minimal impact. Aboveground plant removal is standard for isolating soil GHG fluxes, though belowground autotrophic respiration from adjacent roots remained, as trenching was avoided to reduce site disturbance. Without a control plot, the direct effect of disturbance on GHG fluxes remains uncertain."

- Temperature measurements inside the chamber (reply by authors, last reply on page 19). I agree with the authors that in this case significant effects of a temperature rise in the chambers are unlikely. However, the conversion from ppm to moles will still be improved by adding a temperature sensor.

Agree: Yes, we agree with this and in our new deployments of the SkyLine2D system we are measuring the headspace temperature.

**Report 2 by Daniel Epron**
The authors either addressed almost all of my minor comments or explained why they didn't. While they agree with my major comment, they did not consider it, and I understand their reasoning. In both cases, bias can result from maintaining falsely low values or suppressing truly low values. However, since the concentrations of three gases were measured, some values related to chamber malfunction could have been detected and not confused with a truly low flux, as it would have affected all three fluxes [I agree that it can happen that the three values are truly low].

At least, rather than deleting the result from the database, it would have been possible to flag the questionable flux values and leave future users free to use their own criteria.

Author: Thank you for your comment and understanding our reasoning. We fully agree that flagging questionable fluxes are paramount for the user, regardless of the purpose with using the data. As per request of one of the reviewers we added the standard error of the slope at time zero for each flux in the dataset. This provides the user with an added possibility of flagging fluxes where the fit is uncertain. As described in Section 2.7 the fluxes included already underwent quality control screening using the criteria published in Rheault et al. (2024) and in case of high SE (indicative of overfitting) which is more frequent at low fluxes the linear model was chosen. This approach we consider as the most conservative approach without penalizing the data set using the $R^2$ or flux value per se. However, with this addition of the SE of the slope we hope we have added even more transparency to the quality of the data set.

**Report 3 by Judith Vogt**

I would like to thank the authors for their work on the manuscript. I am still struggling with the fact that this manuscript is meant to be a data paper. It seems that there is a lot to explore, but the authors refrain from doing so to stay within the scope of the journal. I do see that there is a lot of data in this product that could potentially be used by the research community. The authors could be clearer in convincing the reader in this regard. Therefore, I think this paper should be further improved before publication.

Below are more general and some specific comments.

Dataset:

The csv files need revision. The first row should include the header and there should not be any empty or redundant columns. Otherwise, potential users might refrain from using the data if they have to clean it up themselves first. The files still seem to have multiple TIMESTAMP columns and it remains unclear to me why. A Readme file clearly explaining the meaning of column names would be appreciated, e.g., what does Head_05 mean?

Author: Thank for the recommendations. We have gone through all the files to remove redundant columns which was present in the file "VB groundwater depth Figure 6.csv" now "VB groundwater depth Figure 5.csv". In this file we have reduced number of columns and included a "collar" column. This now means that there are five columns in total without the ambiguous numbering. The file "VB SkyLine2D transect Figures 3.csv" has also been revised to only include position data. In the data repository there is a file called "Data variable explanation.pdf" where all variables are explained in table format for each of the csv-files.

Abstract:

The second and third paragraph of the abstract showcase essentially a summary of what I would expect in a Results and discussion section. I think those could be merged and shortened. Afterwards, it would be more valuable to clarify why the presented dataset is useful to the research community. Which processes or relationships could be investigated with this dataset? E.g., those that the authors refrained from further investigating in this paper.

Author: We agree with the suggestion here and have reduced the summary of data to mostly summarizing average values for CO2, N2O and CH4. We have added a new paragraph outlining broadly the opportunities we see with this dataset. This is not an exhaustive list, as we hope that potential future users can find other innovative ways for the data.

l. 11: remove "here" and add "dataset of automated [measurements of] greenhouse gas…"

Author: done

l. 21: mean and SE should be given with the same precision Author: We are unsure how to implement this recommendation. What is meant by precision? We have strived to have equal number and significant digits for average and SE values and view that the numbers we show are detailed enough to provide the reader with a sense of the magnitude and variation of the mean, without being too precise in terms of more or less decimals.

l. 26: split up long sentence Author: This part of the abstract has been reformulated

l. 36: maybe rephrase the sentence too because it is currently confusing – how do CH4 emission bursts have little seasonal variability? Due to their abrupt nature, variability is large both spatially and temporally. Author: Agree that this sentence was confusing and we deleted the last part of the sentence: ", confirming…"

Introduction:

The short discussion on pros and cons of measurement techniques (l. 48-53) could be reconsidered: What is meant by extreme events? Please add an example. Also, I think it is possible to determine seasonal dynamics with chamber-based measurements. Their caveat is rather the spatial component and the required workforce when aiming at high temporal resolution (for manual chamber measurements). Regarding eddy covariance measurements, I think it might be a matter of perspective whether the spatial coverage (on a global scale?) can be called "poor" – it depends on the goal. The authors are aimed at introducing automated measurement techniques here, so adjusting the wording of this paragraph may resolve this controversy.

Author: Yes, we see that this line of argumentation may have been drawn up to sharply and have largely reformulated parts of the paragraph in line 52-63.

l. 44: add "net sources [to the atmosphere]" Author: done

l. 48: add "[manual] chamber-based measurements" Author: done

l. 65-75: seems irrelevant to the present study – consider removing Author: agree. This would be more relevant in a paper dealing with flux calculation procedures

l. 76: remove "uniquely" Author: done

l. 79: might be worthwhile mentioning your measurement system here, i.e. "with an automated GHG chamber system [(SkyLine 2D)]" Author: done

Materials and Methods:

I still suggest some restructuring to avoid repetition and improve readability. First, the site is described, then the main part of the study should be described which are the flux measurements, and afterwards, soil, and other measurements can be described. Move subsections 2.2 and 2.1.4 before 2.1.2. And I would move section 2.4 either before or after 2.1.3. Furthermore, I suggest to merge sections 2.1.4 and 2.6.

Author: Thanks you for taking the time to consider this logical progression of the text. We agree with this division and have largely followed these recommendations. After reorganizing the order of the sections in M&M is:

2.1 Site description
2.1.1 Site preparation and disturbance
2.2 Overview of time series…
2.3 The SkyLine2D system at Vejrumbro
2.3.1 Greenhouse gas flux measurements with the SkyLine2D system
2.4 Peat and organic soil characteristics
2.5 Groundwater table level, depth and sampling
2.5.1 Groundwater water sampling and chemical analysis

Reorganizing like this also meant changing table 1 → 2 and table 2 → 1.

l. 94: change title to "Site description" only Author: done

l. 101: what is meant by "has primarily served as grassland in recent decades due to the wet conditions"? Was the site a wet grassland before drainage in 1950? Or why would the site be wet after drainage? Please clarify/reword. Author: We can see this is an ambiguous formulation it has been changed to the following: "…was used to cut hay for fodder as the conditions were unfavourable for cereal production"

l. 106: Suggest to remove "and disturbance" Author: done

l. 107: remove "Initially" Author: done

l. 115: Suggest to replace "avoiding" with "removing", remove "also", remove "and resolve spatiotemporal patterns to a higher degree than previous studies at this site have achieved and what other commercial platforms are capable of" Author: done

l. 120: Suggest to reword to: "Therefore, we harvested and removed aboveground plants…" Author: done

l. 120-139: there is a lot of repetition in this paragraph, please conflate. Author: this paragraph has been shortened to provide a more concise summary of the harvest and glyphosate application (line 118-130).

l. 145: a reference to current Fig. 3 would be helpful here Author: done

l. 148: Please clarify that effective pH is determined under suspension conditions. Author: done

l. 153-161 and Tab. 1: I would rather expect these in section 3 Data presentation Author: this can also work and these data have been moved to now section 3.6 Peat soil characteristics. Table 1 → Table 3

l. 165: I think by capping the sample, you don't avoid air bubbles, but contamination? Author: done

l. 173: remove "to a 10 mL sample" Author: done

l. 176/Fig. 2: I suggest to merge Fig. 1 and 2 Author: done

l. 181: replace "placed" with "located" Author: done

l. 182: remove "little" Author: done

l. 189 and 194: repetition of collar description Author: text in original line 194 deleted

l. 200: replace "normal" with "common" Author: done

l. 201/Fig. 3: please mark the minimum and maximum of the water depth as shaded areas rather than lines that are not easily distinguishable from the mean Author: done

l. 231: Start phrase with "To measure the depth of the water table, piezometers…" or similar Author: done

l. 241-247: Please make clear here which metric or maybe rather reference point you are using in the data presentation. The abbreviation WTD should also be introduced in this section – is it positive or negative and does it have the ground surface as the reference point? Author: this has now been clarified (line 237-244)

l. 309: "either" should be followed by "or", but it is missing in the sentence. Author: the parenthesis had been placed wrong. Now corrected

l. 311: Is this already accounted for in the numbers given in l. 309? Also, what is meant by "visibly detect" – which criteria was used here to remove low fluxes, also RMSE? Author: we agree with the reviewer that the formulation was ambiguous and we have now clarified that the total number of $CO_2$, $CH_4$ and $N_2O$ fluxes (numbers in line 292) were discarded based on two situations: 1) chamber malfunction and 2) insignificant ($p>0.05$) regression between concentration and time at in situ flux levels close to the minimum detectable flux of the system

l. 318: Do the annual cumulated fluxes here refer only to the diffusive fluxes? Please clarify. Author: we have now added that for the ditch in this section that it was only the diffusive fluxes. The annual upscaling for ebullition is described in line 327-329. For the peat soil we did not observe ebullition

l. 338: refine "classified as ebullition events" Author: the classification we used is described in line 308-312 and example shown in Fig. S4. We now refer in the text to this classification

l. 343: would it not be handier to convert the ebullition flux in the dataset to more common units (per second instead of per 5 min)? Author: This is a good suggestion and in the data file we have now converted the ebullition fluxes to s-1 to be compatible with the rest of the fluxes. This change in units also result in that the text originally in line 341-346: "Furthermore, ebullition flux is calculated as…" has been deleted.

l. 344: why not use precise numbers here – i.e. 19.3% and 365 days per year? Even if 360 days were measured, 365 days make the annual estimate. Author: there was a mistake in the number we initially added and the correct is now added (line 327-329).

l. 347: shouldn't it be 60 s instead of 30 s, or are you not referring to the delay time of 60 s? Author: Yes. This has now been corrected to 60 seconds. Was a leftover from earlier discussions on this.

Data presentation:

l. 353: worth mentioning the time period here once more Author: done

l. 359/Fig. 4: consider using a color palette that is easier to distinguish Author: done

l. 365: Revise caption: Does collar 4 have soil temperature measurements or not? – Tab. 2 states yes, Methods states no, Fig. 3 states no; "along the measurement" probably means "along the transect"; add "blue dots are the raw 5 min measurements of air [and soil] temperature" Author: Thank you for spotting this error. We have now added in M&M section 2.6 (line 259-263) that we initially inserted soil moisture/temp sensors at collar 1, 4, 7, 9, 18, 23 and 27, but sensor failures limited this coverage. This has also been updated in the current Fig. 2 showing the transect.

l. 369: reference to Tab. 2 does not seem correct, probably Tab. 3 is meant. Please double-check the referencing of figures and tables. Author: after rearranging the sections the reference to table 2 is now correct. We have double checked all references to tables and figures after the reorganization of sections and results etc.

l. 388: Unit missing "below -40" and now the reference point seems to a different one because of the negative number. Please be consistent throughout. Author: Corrected to "40 cm" and yes negative representation is wrong and we have checked throughout.

l. 389: add "the deepest groundwater [table]" and do you mean "transect" instead of "site"? Author: corrected

l. 401: Please update the caption, there are no green lines. Author: done

l. 411: What is meant by "drying properties of the soil"? Maybe reword. Author: we have now reformulated to indicate that the contrasting modes of decreasing soil moisture may be because of different water retention properties.

l. 424: Please revise the sentence. Author: We have now reformulated the beginning of the sentence to "This temporal trend was also observed…"

l. 438: remove "especially elements related to" Author: done

l. 439: "dynamic" is a very vague word. Please be a bit more concise what you mean. Author: we have reformulated to: "…where the chemical composition of groundwater varied more over time than ditch water."

l. 445: I am unsure whether it is clear what you mean by site. I would assume you mean the whole transect. Ideally define this already in the Methods section. Author: we have gone through the entire manuscript and corrected where it is more appropriate to mention "transect" rather than "site". Furthermore, we use "site" together with "Vejrumbro" to indicate a broader

geographical location rather than the specific transect.

l. 467-471: adding references here would be appreciated Author: this part of the text (line 464-470) has been changed and a couple of relevant papers have been cited.

l. 473: Why "drained"? Do you refer to the past drainage at the site? Author: drained has been deleted

l. 489: I think GWP might have been confused with GHG? Author: yes and no. We actually mean the global warming potential (GWP), but we have reformulated to "…GHG budget in relation to the global warming potential"

l. 502/Fig. 10: Some figures have month axes with abbreviations (Fig. 10), some with numbers (Fig. 12). Please unify. Author: all figures now have month in numbers on the x-axis with year of measurement also added where relevant

l. 504: correct ST to Tsoil Author: done

l. 554: "Net uptake" of what? Author: corrected to "Net uptake of $CH_4$…"

l. 573: replace "diffusive" with "diffusion" and again in l. 606 Author: done

l. 575: I find "varies most throughout the measurement period" more precise than "is most dynamic" Author: done

l. 632-635 This section appears out of nowhere. It mentions some methodological hints that should be inserted in the Methods section. If annual estimates are calculated, please state the numbers here or in a table. Were the budgets converted to GWP? If so, which time frame was used? Author: section has been deleted and was by mistake left over from the previous revision. All cumulative fluxes is mentioned now in sections 3.8.1 and 3.8.3.1 (ditch). The simple upscaling to annual estimates is outlined in lines 304-309 (section 2.8) and 330-332 (section 2.9).

Data availability:

The link to the dataset directs to "Page not found". Author: sorry for this confusion. It was an older link and the correct one - https://doi.org/10.60612/DATADK/BZQ8JE - has now been added to the Data Availability section

Conclusion:

I am having difficulties with some of the conclusions since they are – to my understanding – based on visual inspection and not statistical analyses or any in-depth analyses of the data. Author: We are unsure which specific parts the reviewer refers to. As mentioned in the first review the purpose of this data descriptor paper is for objective and simple presentation of the data and not hypothesis driven research that would necessitate the use of specific and diverse types of statistical analysis depending on the hypothesis being tested.

We have rearranged the conclusion and attempted to be more cautious in the wording, so as not to suggest that we did statistical test. However, in the conclusion regarding we do write how that there is a link to thermal and hydrological drivers are linked to GHG, but in more general terms. These links are well established knowledge, based on decades of research in peatlands. We hardly see a need to make formal statistical tests demonstrating to prove this again. The very dynamic nature of the data also beckons the question which types of statistical analyses are then most suited to test this. Simple linear regression could be one, and we agree with the reviewer that this would have provided a quantitative basis for the conclusion, but not necessarily providing a deeper insight to unravel short term temporal trends that may be more non-linear.

Below we have highlighted the conclusions we have changed to reflect the observatory nature of data interpretation rather than the use of more statistics:

1) "However, spatial variation of cumulative fluxes for all GHG were not directly related to WTD levels, contradicting the general assumption that WTD is the primary driver of GHG

emissions." We have changed the wording of this conclusion to "Specifically, the dataset demonstrates how temporal variation in soil hydrology and temperature is linked to the temporal variation of fluxes. Interestingly, the temporal variability of GHG fluxes across the transect appears to be lower than the spatial variation highlighting that spatial variability in hydrology and temperature may not necessarily be the best predictor of flux magnitudes across the transect." (lines 639-642) and have deleted the word directly, so as not to suggest a tested relation, but rather communicate that it is based on observation.

2) "Cumulative soil N2O fluxes exceed what has been previously reported for temperate fens, but show similar seasonal regulation by ST" have been changed to ": Cumulative soil N2O fluxes exceed previously reported values for temperate fens at the Vejrumbro site and others. Unlike CO2, N2O is emitted largely in pulses related to rapid fluctuations of WTD, which increase in size with Tsoil, indicating a seasonal regulation of N2O production by temperature." (lines 656-658)

3) "A likely cause for the high soil N2O emissions could be a combination of leaching of inorganic nitrogen from surrounding agricultural fields and release of organic N from the decomposing peat." this has been deleted from the conclusion.

4) "The site was during the measurement period an insignificant source of soil CH4, which is likely due to the well-drained summer period, a cold wet winter and presence of the major electron acceptors (NO3-, SO42- and Fe3+), providing suboptimal conditions for CH4 production." has been changed to "The peat soils across the transect were insignificant sources of soil CH4 during the measurement period. This could be linked to deeper WTD (20 – 40 cm) during the summer period, a cold wet winter and presence of alternative electron acceptors (NO3-, SO42- and Fe3+), which provide suboptimal conditions for CH4 production." (lines 662-664). We argue that keeping the mentioning of the possible impact of the alternative electron acceptors is warranted as we have now added a reference stating this in the discussion (Bridgham et al. line 467) and that this is so well known regarding CH4 regulation in peatlands that it does not require a statistical test. It would be possible to make more in depth analysis of the CH4 fluxes – geochemistry relation for the single piezometers and collars, but then again what we communicate here is more of an indirect effect of the electron acceptors, as the most direct effect of these in relation to CH4 would be expected to be the redox potential.

---

## Author Response (AR3)

Comments to review

Thanks for the comments

1) Checking your paper, I noticed that your Table 2 contains coloured cells. Please note that this will not be possible in the final revised version of the paper due to HTML conversion of the paper. When revising the final version, you can use footnotes or italic/bold font. For now, the process will continue, but please note that the final version cannot be published by using coloured tables. Author: We assume you refer to the current table 1 showing data availability. This is the only table containing coloured cells. We have removed the coloured cells and replaced them with horizontal bars. In addition we streamlined the format of the other tables.

2) Please ensure that the colour schemes used in your maps and charts allow readers with colour vision deficiencies to correctly interpret your findings. Please check your figures using the Coblis – Color Blindness Simulator (https://www.color-blindness.com/coblis-color-blindness-simulator/) and revise the colour schemes accordingly. --> Figs. 3 Author: regarding figure 3. We checked in the advised simulator for all possible settings. We assess that it appears that all colors are represented clearly distinguishable and keep it at it is shown now. For all the other figures except fig. 1 and 2, each panel is given with a heading indicating the gas shown on that specific panel. This will aid in the interpretation of the figure and also in the case the figures were shown in grey/black/white. We therefore assume these figures to be adequate for all readers.